# Capturing the temporal evolution of choice across prefrontal cortex

**Laurence T Hunt[1,2]\***, **Timothy EJ Behrens[2,3]**, **Takayuki Hosokawa[4,5,6]**,
**Jonathan D Wallis[4,5]**, **Steven W Kennerley[1,4,5]**

[1]Sobell Department of Motor Neuroscience, University College London, London,
United Kingdom; [2]Wellcome Trust Centre for Neuroimaging, University College
London, London, United Kingdom; [3]Oxford Centre for Functional MRI of the Brain,
Nuffield Department of Clinical Neuroscience, Oxford University, John Radcliffe
Hospital, Oxford, United Kingdom; [4]Helen Wills Neuroscience Institute, University
of California, Berkeley, Berkeley, United States; [5]Department of Psychology,
University of California, Berkeley, Berkeley, United States; [6]Laboratory of Systems
Neuroscience, Graduate School of Life Sciences, Tohoku University, Sendai, Japan

**Abstract** Activity in prefrontal cortex (PFC) has been richly described using economic models of
choice. Yet such descriptions fail to capture the dynamics of decision formation. Describing
dynamic neural processes has proven challenging due to the problem of indexing the internal state
of PFC and its trial-by-trial variation. Using primate neurophysiology and human
magnetoencephalography, we here recover a single-trial index of PFC internal states from multiple
simultaneously recorded PFC subregions. This index can explain the origins of neural
representations of economic variables in PFC. It describes the relationship between neural
dynamics and behaviour in both human and monkey PFC, directly bridging between human
neuroimaging data and underlying neuronal activity. Moreover, it reveals a functionally dissociable
interaction between orbitofrontal cortex, anterior cingulate cortex and dorsolateral PFC in guiding
cost-benefit decisions. We cast our observations in terms of a recurrent neural network model of
choice, providing formal links to mechanistic dynamical accounts of decision-making.

**\*For correspondence:** laurence.
hunt@ucl.ac.uk

**Competing interest:** See
page 21

**Reviewing editor:** Michael J
Frank, Brown University, United
States

## Introduction

Correlates of decision variables are routinely found in prefrontal cortex (PFC) during value-guided
decision making (*Clithero and Rangel, 2014*; *Kennerley and Walton, 2011*). They have been richly
described using static, economic models of choice (*Glimcher and Fehr, 2014*). Neuroeconomic
accounts explain firing rates of single neurons or human neuroimaging data in terms of experimental
variables that motivate choice behaviour. These include the magnitude or likelihood of available
reward, and the costs involved in obtaining those rewards. By forming neural representations of
such quantities, it is argued that the brain can use these representations to guide selection of the
most valuable alternative. Our present understanding of economic decision formation has been
founded upon careful study of neural representations of value and how they differ across PFC subre-
gions (*Glimcher and Fehr, 2014*; *Rushworth et al., 2012*; *Rangel and Hare, 2010*).

However, an alternative perspective on the origins of neural representations during decision mak-
ing stems from choice models from mathematical psychology (*Busemeyer and Townsend, 1993*).
Such models seek to explain decisions in terms of their temporal evolution or *dynamics*. Whilst origi-
nally accounting for temporally varying features of behaviour, such as reaction times and eye gaze
(*Busemeyer and Townsend, 1993*; *Ratcliff and Rouder, 1998*; *Krajbich et al., 2010*), they have
also successfully captured temporally varying features of neural activity, most notably in the lateral

**eLife digest** In 1848, a railroad worker named Phineas Gage suffered an accident that was to secure him a place in neuroscience lore. While constructing a new railway line, a mistimed explosion propelled an iron bar into the base of his skull, where it passed behind his left eye before exiting through the top of his head. Gage survived the accident, but those who knew him reported significant changes in his personality and behaviour.

Gage's ability to make decisions was particularly impaired by his injury. Decision-making involves weighing up the costs and benefits associated with alternative courses of action. It entails looking into the future to decide whether an anticipated reward will justify the effort or expense necessary to obtain it. This process is dependent on a region of the brain called the prefrontal cortex, the area that sustained the most damage in Phineas Gage.

While many studies have shown correlations between activity in particular parts of prefrontal cortex and the outcome of decisions, little is known about how this activity evolves over time as a decision is made. To explore this process, Hunt et al. trained macaque monkeys to choose between pairs of images that were associated with specific rewards (quantities of fruit juice) and costs (either amounts of work or fixed delays).

Electrode recordings revealed changes in prefrontal activity that varied over time as the monkeys deliberated over each pair of images, choosing for example between a large reward after a long delay versus a smaller reward immediately. This activity was consistent with a mathematical model of decision-making, which also explains data from brain imaging experiments in humans. This provides an important link between human data and electrode recordings in animals.

However, some of the patterns of activity observed in both macaques and humans appeared to reflect the speed at which decisions were made, rather than the outcome of the decisions themselves. By extracting information about decision speed on each decision from each region, it was shown that communication between regions of prefrontal cortex changes when choices are between two different amounts of work, as opposed to two different delays. Further experiments are needed to explore this phenomenon and to determine how other brain regions interact with the prefrontal cortex to support the decision-making process.

intraparietal cortex (LIP) during perceptual choice (*Shadlen and Newsome, 2001*; *Gold and Shadlen, 2007*). These psychological models can also be related to other dynamical models (*Bogacz et al., 2006*) such as nonlinear attractor network models firmly rooted in neurobiology (*Wang, 2002*). The common feature of both psychological and neurobiological accounts is that they focus on the temporal evolution of a decision signal *across* time, rather than the strength with which it represents decision variables at a particular fixed point *in* time.

Contrasting the two perspectives, it becomes apparent that even if neuronal activity underwent the exact same physiological process on different trials (such as a ramp-to-threshold), it might nevertheless appear to represent certain economic features of a decision. Put simply, as the decision unfolds, neural activity will be higher on trials that ramp quickly than on those that ramp slowly. It will therefore correlate with any variable that affects the decision speed, and this effect will appear most prominent at timepoints in the trial when the ramping process, or rate of change, is maximal. Even in more complex dynamical situations, it is easy to see how economic variables that predict the rate of change would appear to be represented in neural activity if analysed without knowledge of the underlying dynamics. This might particularly be true of the value of the chosen item ('chosen value'), which strongly affects behavioural reaction times in economic choice tasks.

This should then influence how we interpret the meaning of such activity (*Hunt, 2014*; *O'Doherty, 2014*). The neuroeconomic perspective often labels chosen value representations as a 'post-decision' signal, arguing that chosen value signals do not reflect the decision competition itself but instead the outcome of the comparison process (*Cai and Padoa-Schioppa, 2012*; *Blanchard and Hayden, 2014*). One interpretation of such representations is that they are needed for subsequent computation of a reward prediction error (*Rangel and Hare, 2010*). Yet the dynamical perspective proposes that they may in fact originate as a consequence of time evolving decision processes. Rather than casting a chosen value representation as signaling 'pre-decision' or 'post-decision'

variables to downstream brain areas, it argues that such correlates inevitably emerge *as* a decision is made. In its most extreme form, this hypothesis might contend that chosen value signals would be fully accounted for by variation in underlying decision rates.

To assess this proposal more carefully, it becomes critical to index how neural dynamics unfold on individual trials. In PFC, tackling this problem provides unique challenges. Several recent studies have recovered single-trial dynamical information in structures close to motor output, based upon neuronal spiking data (*Thura and Cisek, 2014*; *Kaufman et al., 2015*; *Murakami et al., 2014*; *Bollimunta et al., 2012*; *Kiani et al., 2014*; *Carnevale et al., 2015*). Yet the distance of PFC from either sensory input or motor output produces neural activity that is poorly aligned with simple features of the experimental task (*Rigotti et al., 2013*). Attempts to understand PFC activity should therefore extract its dynamical state based on neural information alone, rather than first sorting spike trains by motor output or experimental variables as is common in other approaches. It is also unclear whether neuronal firing rates are in fact the most reliable source of single-trial information. The esotericism of PFC neuronal responses is compounded by a high degree of stochasticity in neuronal spike trains, delivering statistical challenges to obtaining single-trial information (*Churchland et al., 2007*; *Park et al., 2014*). Current techniques sometimes overcome this problem by examining tens or hundreds of simultaneously recorded single neurons (*Churchland et al., 2007*). Whilst this can be fruitful, such data is rarely available in humans, where the contribution of PFC to value-guided choice is most critical.

In the present study, we therefore sought an alternative index of single-trial neural dynamics that overcame these limitations. We focussed on observations at the mesoscopic scale of the local field potential (LFP). One advantage of this approach is that it can be easily related to simultaneously recorded neuronal spike trains on a trial-by-trial basis in animals, but also underlies the magnetoencephalography (MEG) signal observable in humans (*Buzsáki et al., 2012*). As such, we could recover dynamical information from electrophysiology recordings in macaque monkeys and also from non-invasive magnetoencephalography (MEG) recordings in humans. This provides an important bridge between observations at microscopic (cellular) and macroscopic (whole-brain) scales. Importantly, our index provides information about the speed of a *local* internal neural decision process that goes beyond a simple behavioural measure of reaction time.

Using this index, we could draw several new conclusions about PFC activity during choice. We first demonstrate a temporal evolution from action value difference to chosen action signals that are selectively present in dorsolateral prefrontal cortex (DLPFC) neurons, but not anterior cingulate cortex (ACC) or orbitofrontal cortex (OFC). This contrasts against the ubiquitous encoding across all three subregions of 'chosen value'. However, we then demonstrate that correlates of chosen value may arise as a consequence of dynamical processing, as opposed to being purely represented as a neuroeconomic post-decision variable. Next, by simultaneously indexing decision formation across multiple PFC subregions, we show that functional interactions can be reshaped in a task-dependent manner. We find that OFC and ACC dynamics selectively influence dorsolateral DLPFC activity on delay- and effort-based decisions respectively. Finally, our observations can be related to predictions from a dynamical neural network model of choice. This provides formal links to established dynamical mechanisms underlying perceptual decision-making.

## Results

We examined a study of cost-benefit decision making in which single neuron firing and local field potentials (LFPs) were recorded simultaneously from three PFC subregions fundamental to value-guided choice in four macaque monkeys: orbitofrontal cortex (OFC), anterior cingulate cortex (ACC) and dorsolateral prefrontal cortex (DLPFC). On each trial, subjects chose between two pictures that led to a certain reward magnitude (quantity of fruit juice) after paying a certain cost (*Figure 1A*). On half of trials this cost was physical effort needed to obtain reward, and on half of trials it was delay to reward. Subjects were overtrained on the cost and benefit paired with each picture. Their choices were well described by a linear trade-off between reward and cost combined with a softmax choice function (see *Hosokawa et al., 2013* for detailed behavioural modelling). This allows a straightforward definition of 'value' used in the analyses below (see methods). Throughout the paper, we analyse neuronal firing and LFP during a 1s choice phase, when subjects fixated centrally whilst both

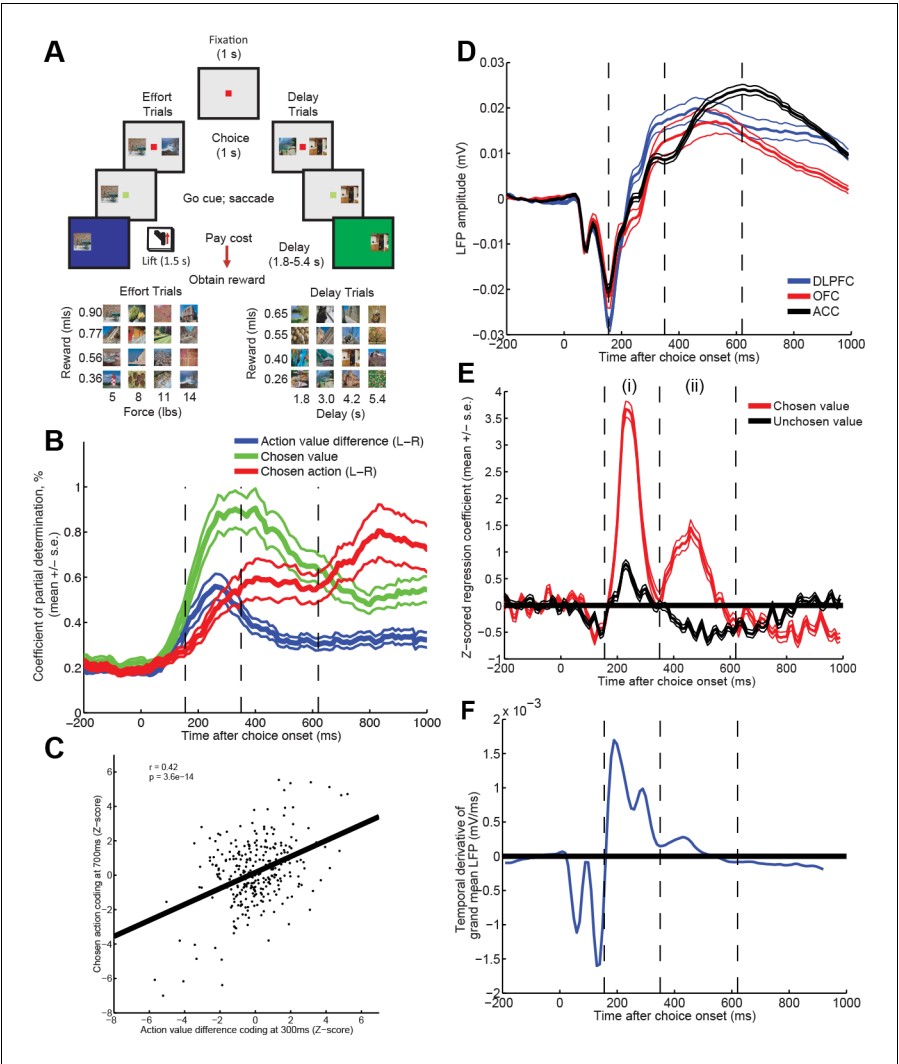

**Figure 1.** Time-varying value correlates in single units and LFP during choice. (A) Subjects chose between two pictures of differing value (reward and cost (physical effort, or delay)) by saccade. Neural activity was examined during the 1s choice epoch, whilst monkeys held fixation. (B) Correlates of decision variables in DLPFC single units (n=303). Coefficient of partial determination for multiple regression of left minus right action value (blue), chosen value (green) and left minus right choice (red) onto single neuron activity. Lines show mean /- s.e. across all recorded neurons. See also *Figure 1—figure supplement 1*. (C) Early action value difference coding predicts late chosen action coding. Z-scored regression coefficients for left minus right action value at 300 ms (ordinate) are plotted against Z-scored regression coefficients for left minus right choice at 700 ms (abscissa), for each DLPFC neuron (R=0.42; p=3.6*10$^{-14}$). (D) Baseline-corrected ERP during choice epoch, in example subject, split by region. Lines denote mean /- s.e. across n = 125 (DLPFC)/n = 43 (OFC)/n = 85 (ACC) electrodes. Other subjects plotted in *Figure 1—figure supplement 2*. Vertical dashed lines are added to allow latency comparison with (E) and (F). (E) Z-statistic of regression of LFP data from example subject onto chosen and unchosen value. Note that at timepoint (i), overall value =chosen+unchosen; at timepoint (ii), value difference = chosen-unchosen. Lines denote mean /- s.e. across electrodes. In *Figure 1—figure supplement 3*, figure is split into different brain regions and cost/benefit. (F) Temporal derivative of evoked potential in part (D) (averaged across regions). Comparing parts (D), (E) and (F) shows that value correlates at timepoints (i) and (ii) occur when the LFP is ramping (derivative is non-zero), rather than peaking (derivative is near or at zero).

The following figure supplements are available for figure 1:

**Figure supplement 1.** Correlates of decision variables in DLPFC, OFC and ACC single units, as in *Figure 1B*.

**Figure supplement 2.** Consistency of evoked potentials across subjects, and brain regions, in macaque PFC.

**Figure supplement 3.** Effects of decision variables across different brain areas and reward vs. cost.

pictures were on screen. After this 1s period, a go cue appeared, instructing subjects to saccade to their preferred picture to indicate their choice.

## Value correlates vary across time in both single units and LFPs

We first adopted a classical approach to analysing our data, examining features of the task represented by neural activity. Using multiple regression on single neuron firing rates, we observed a dynamic (time-varying) signature of value correlates in the DLPFC (n=303 neurons) (*Figure 1B*). Early in the trial, value correlates were found in the reference frame of *action value difference* (i.e. left minus right saccade value influenced neural firing; *Figure 1B*, blue), but late in the trial these evolved into correlates of the eventual *chosen saccade* (left vs. right chosen saccade influenced neural firing; *Figure 1B*, red) (*Kim et al., 2008*; *Louie and Glimcher, 2010*). DLPFC neurons that showed strong selectivity for left minus right saccade option value 300 ms after decision onset also showed strong selectivity for left minus right choices 700 ms after decision onset (*Figure 1C*). This temporal evolution from action value difference to categorical choice was selectively present in DLPFC, but not ACC (n = 321 neurons) or OFC (n = 212 neurons) (*Figure 1—figure supplement 1*). Notably, however, all three regions showed correlates of *chosen value* with a similar timecourse (*Figure 1B*, *Figure 1—figure supplement 1*).

We then applied the same approach to consider dynamics of LFP value correlates (electrodes: DLPFC, n=208; ACC, n=207; OFC, n=146). The overall shape of the choice phase evoked LFP was surprisingly consistent across PFC subregions (*Figure 1D*) and subjects (*Figure 1—figure supplement 2*). All areas contained a fast biphasic component shortly after sensory input, and a slower component lasting several hundred milliseconds after this. Because of this similarity, we collapsed across regions (see *Figure 1—figure supplement 3* for regions separately), and regressed LFP amplitude onto both *chosen value* and *unchosen value* across time. This revealed two timepoints at which correlates of value emerged (*Figure 1E*). Early in the trial (~250 ms) value correlates were positive for both chosen and unchosen value, but later in the trial (~450 ms) value correlates for unchosen value flipped to become negatively signed. This implies a temporal evolution from an early representation of value sum (=chosen+unchosen value) to a later representation of value difference (=chosen-unchosen value). This progression is notably similar to observations made in human PFC during value-guided choice using MEG (*Hunt et al., 2012*). Crucially, these value correlates were maximal when the event-related LFP was *ramping*, not *peaking* (that is, when the temporal derivative of the LFP was large [*Figure 1F*]). This suggests chosen and unchosen value influence the rate at which LFP dynamics unfold on each trial, rather than being explicitly represented by the amplitude of an evoked response.

As revealed by these classical analyses, value correlates vary across time at both microscopic (single unit) and mesoscopic (LFP) scales. Yet it remains unclear how we might bridge these dynamics at different scales. Below, we demonstrate that forming such bridges is of fundamental importance, especially if it can be achieved at the level of individual trials. It allows us to interrogate the relationship between neural dynamics and behaviour, the encoding of neuroeconomic variables, and how brain regions interact during choice. Moreover, by applying to the same approach to human data, it links observations in human MEG recordings with their underlying neuronal counterparts.

## Extracting single-trial information about choice dynamics from event-related LFP

One approach to bridging across data scales would be to extract single-trial information about choice-related dynamics from one scale, and use it to explain variance in the other. The evoked LFP appeared consistent across different recording electrodes and sessions (*Figure 2—figure supplement 1*), and possessed a high single-trial signal to noise ratio (*Figure 2—figure supplement 2*). We therefore pursued a data-driven index of single-trial dynamics from LFP data.

The approach we adopted was to apply principal components analysis (PCA) across trials (see Methods). For each subject, the input data for macaque PCA was decision-locked raw LFP data, stacked across all electrodes from all recording sessions to form a large matrix **X**. **X** has dimensions nSingleTrials by nTimebins (*Figure 2A*). PCA decomposes **X** into temporal principal components **V**, with dimensions nTimepoints by nComponents, and component weights **U**, with dimensions nSingleTrials by nComponents (*Figure 2A*). The principal components (PCs) in **V** provide a set of temporal

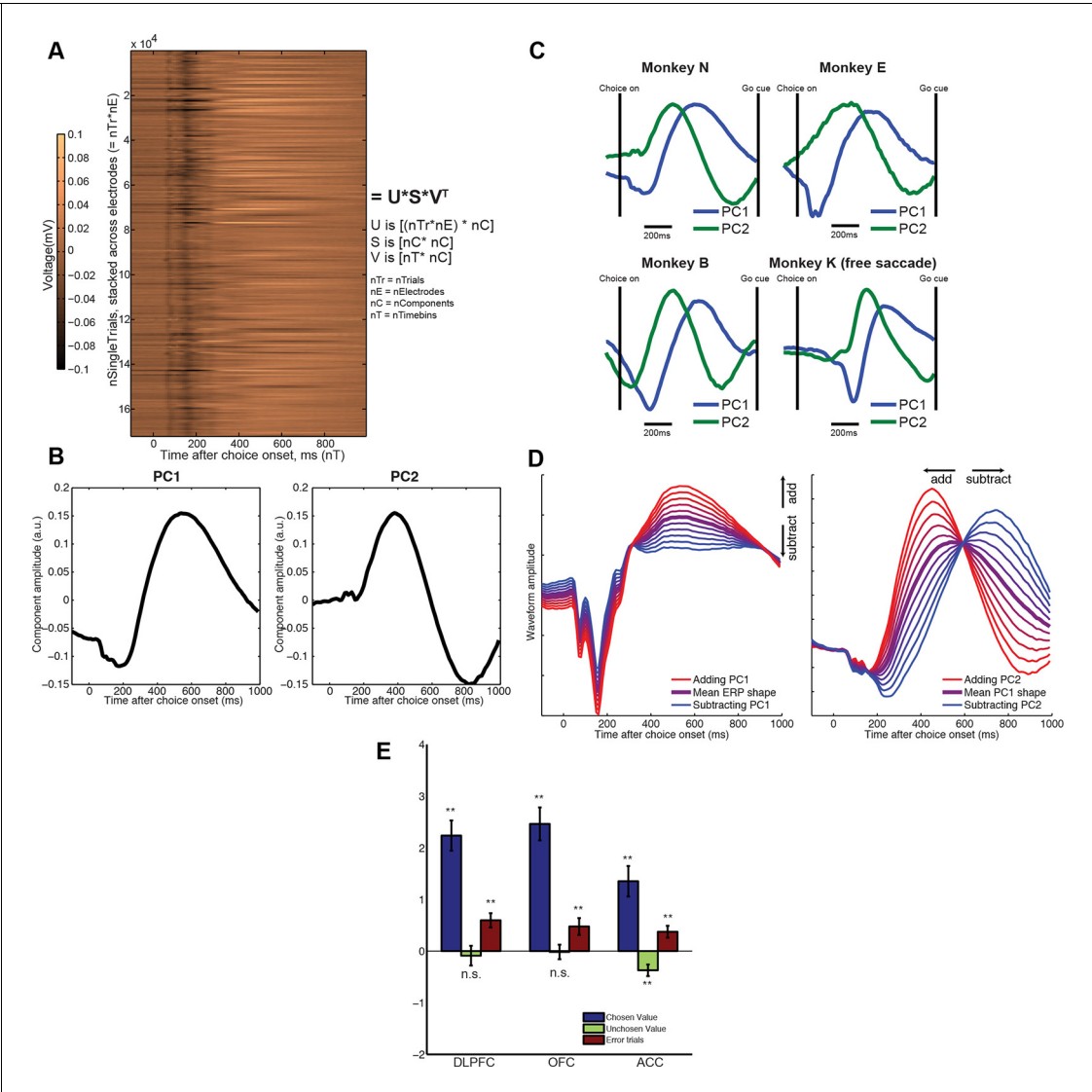

**Figure 2.** Extraction of internal dynamics from LFP data via principal components analysis (PCA). (**A**) A large matrix of single-trial data is formed by stacking data across all recordings. This was performed separately within each subject. Consistent evoked potentials were found across recording electrodes (*Figure 2—figure supplement 1*) and trials (*Figure 2—figure supplement 2*). The single trial-weights **U** are returned separately for each trial on each electrode, but their interpretation is determined by the shape of the components in **V**, and so is common to all electrodes. (**B**) The first and second principal components of this matrix, for example macaque subject N. (**C**) The shapes of PC1 and PC2 are consistent across subjects. (**D**) Left panel: the effect of adding/subtracting PC1 to the main ERP shape in example subject N, modulating its amplitude; right panel: the effect of adding/subtracting PC2 to PC1, modulating its latency. As shown in *Figure 2—figure supplement 3*, this shift in latency can also be related to changes in low-frequency oscillatory phase during the decision period. (**E**) The influence of chosen value, unchosen value and error trials on PC2 scores (i.e. the second column of matrix **V**), estimated via multiple regression. Bars show regression coefficients (a.u.), mean +/- s.e. across electrodes (macaque). ** denotes p<0.01, one-sample T-test. Effects are shown separately for monkey K (who saccaded freely during choice epoch, and indicated response using joystick) in *Figure 2—figure supplement 4*. Influence of variables on PC1 scores shown in *Figure 2—figure supplement 5*.

The following figure supplements are available for figure 2:

**Figure supplement 1.** Consistency of evoked potentials across different recording electrodes/recording sessions, in an example subject (monkey N).

**Figure supplement 2.** High signal to noise ratio in single-trial LFP responses.

**Figure supplement 3.** The relationship between single-trial PC2 weights and single-trial time-frequency decomposition (estimated across all electrodes in DLPFC).

*Figure 2. continued on next page*

*Figure 2. Continued*

**Figure supplement 4.** Regression of decision variables onto PC2 (left) and PC1 (right) in monkey K (free saccade).

**Figure supplement 5.** Regression of decision variables onto PC1.

basis functions (*Figure 2B*) that capture the principal modes of variation in the shape of the waveform across trials. The single-trial weights in **U** tell us how much of each PC is present in each single-trial response. They are returned separately for each electrode. Because data are stacked, however, the decomposition has the same meaning across different electrodes and recording sessions.

The top two PCs were strikingly consistent across all four macaque subjects (*Figure 2C*). Importantly, they were readily interpretable in terms of their effects on LFP dynamics. PC1 (*Figure 2B*, left panel) resembled the basic shape of the event-related field potential (*Figure 1D*). Adding or subtracting PC1 therefore captures variation across trials in response *amplitude* (*Figure 2D*, left panel). More notable, however, was PC2, which was relatively flat at the beginning of the decision, but from 200 ms onwards resembled the *temporal derivative* of PC1 (*Figure 2B*, right panel). Adding or subtracting the temporal derivative of a waveform controls the *latency* at which it peaks (*Figure 2D*, right panel) (*Friston et al., 1998*; *Mayhew et al., 2006*). Knowledge about PC2 weights therefore provides a parsimonious description of single-trial ERP latencies, a key feature of the dynamical ERP response. Consistent with this idea, PC2 weights were also found to correlate with the phase of low frequency (theta frequency [4-–8 Hz]) oscillations during the decision period (*Figure 2—figure supplement 3*).

In light of this, we hypothesised that factors modulating reaction time in value-based choice studies (*Busemeyer and Townsend, 1993*) would affect the weight of PC2, controlling the latency of the LFP waveform. We found *chosen value* had a strong and consistent effect on PC2 across all regions studied. When chosen value was higher, PC2 was more positive, implying the waveform peaked earlier in time (*Figure 2E*; *Figure 2—figure supplement 4*). On 'error trials', where the subject chose the less valuable option, PC2 weights were more positive than on correct (*Figure 2E*). Under the assumption that higher value trials, and error trials, were associated with faster decision dynamics (*Busemeyer and Townsend, 1993*; *Ratcliff and Rouder, 1998*), PC2 weights therefore provide a neurally-derived index of the speed of the decision on each trial. Crucially, however, they are obtained separately for each individual electrode. They therefore provide a *local* measurement of neural dynamics, beyond a simple behavioural measure of reaction time. PC1 weights, capturing response amplitude rather than latency, were primarily influenced by value sum (same sign for both chosen and unchosen value, with the exception of ACC) (*Figure 2—figure supplement 5*).

## Internal single-trial dynamics explain neural activity over and above external variables

We next considered whether our *internal* LFP-derived index might explain features of neural firing that we could not previously explain using *external* experimentally-derived variables. We explored this idea using multiple regression. We regressed experimental factors onto neural firing rates (see *Supplementary file 1* for full list) to capture variance explained by these factors, as in *Figure 1B*. However, we also included as a coregressor the single-trial PC2 weight for that trial, estimated from the decomposition of the LFP. This allows us to examine the influence of PC2 weights on neuronal firing, having controlled for the contribution of all external decision variables. To avoid contamination between spikes and LFP, we used LFP data recorded simultaneously from a neighbouring electrode in the same cortical area. (Note that reaction time is not included as an additional measure of trial-by-trial decision dynamics in our task, as the imposed 1s choice delay prior to response led to a floor effect in subjects' response times.)

*Figure 3A and B* shows two individual neurons from DLPFC that exemplify the action value-to-choice transformation depicted in *Figure 1B/C*. As can be seen, variance in their firing is explained by the action value difference early (~200-–400 ms), and the chosen action late (~600–800 ms) in the decision process. In both cases, the LFP-derived PC2 single trials weights explain additional variance as this value-to-choice transformation occurs (*Figure 3A/B*). Across the population, PC2 had a similar effect at this point in time (*Figure 3C*). 44.9% of neurons in DLPFC were found to have a significant

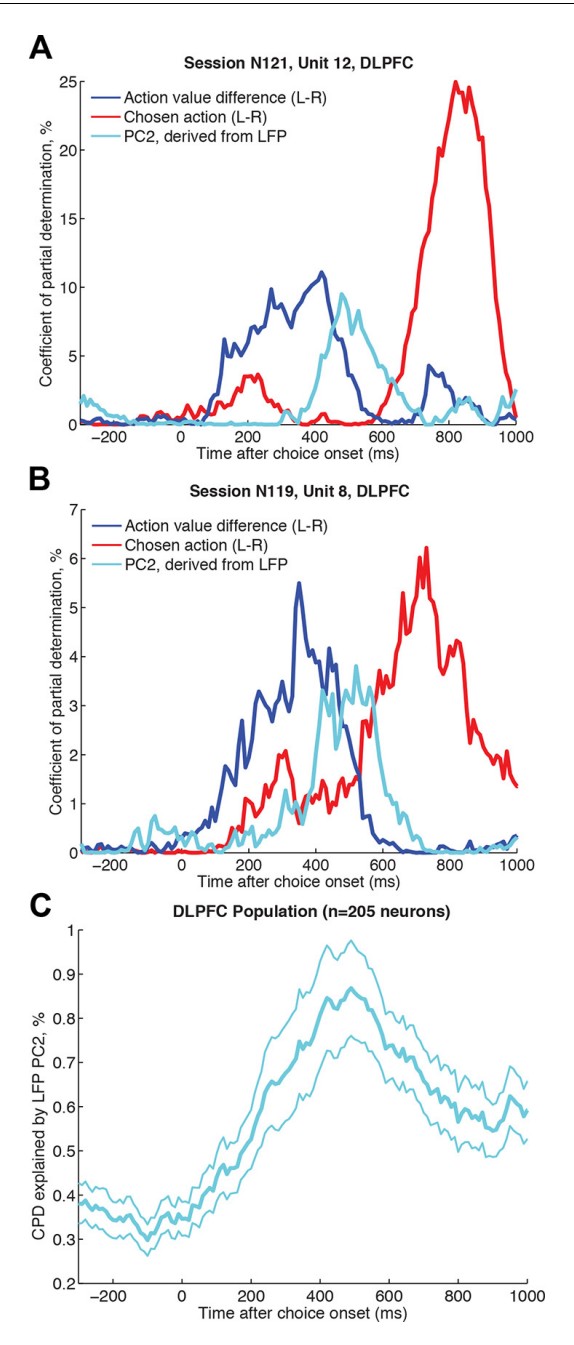

**Figure 3.** Single unit firing in DLPFC explained by simultaneously recorded LFP dynamics, over and above contribution from experimental variables. (**A**)/(**B**) Two example neurons from DLPFC each showing a transition from encoding action value difference (blue) to later encoding the selected action (red), and influenced by the LFP-derived PC2 single trial weights (cyan) in the intervening period. The coefficient of partial determination (CPD) is plotted for all three factors (see Methods for full list of other task-related variables included in regression model). (**C**) Timecourse of population CPD explained by LFP-derived PC2 weights (for n=205 DLPFC neurons that had a simultaneously recorded LFP on a separate DLPFC electrode). Lines show mean /- s.e. across neurons.

modulation by LFP-derived PC2 weights during the choice epoch (23.4% positively modulated, 21.5% negatively modulated, p<0.01 in a 250 ms–750 ms window, corrected for multiple comparisons across time). This finding forms a link between choice dynamics at mesoscopic and microscopic scales, over and above that which can be obtained from examining time-varying value correlates (as

in *Figure 1*). Additional analyses confirmed that this relationship could not be explained as a simple consequence of first-order correlations between LFP amplitude and firing rate (*Whittingstall and Logothetis, 2009*).

Having formed this link between our measure of single-trial LFP dynamics and neural firing, we could then address several questions concerning the roles of these dynamics in value-guided decision making.

## Influence of single-trial LFP components on single unit chosen value correlates

The representation of chosen value is found in many neural structures during choice, but its interpretation has remained unclear. It has been pointed out that this is a 'post-decision' variable, and several suggestions have been offered for why this might need to be encoded (*Rangel and Hare, 2010*). In our study, the timecourse of this signal (found across all three cortical areas, *Figure 1—figure supplement 1*) appears similar to the timecourse of influence of the LFP PC2 weights on neuronal firing (*Figure 3C*). We sought to explore the relationship between chosen value correlates identified in single unit firing and our single-trial indices of ERP amplitude (PC1) and latency (PC2).

To address this, we reanalysed the findings in *Figure 1B*, but asked whether including the top two LFP principal components in our regression model selectively reduced the variance explained by chosen value, but not by action value difference or chosen action (see Methods). As a control, we compared this model to one where two noise components (PC101 and PC102) from the LFP PCA were included as coregressors instead of PC1 and PC2. We found that including the LFP-derived PC1/PC2 as coregressors caused a significant reduction in chosen value coding, but not of action value difference or chosen action coding (*Figure 4*; *Figure 4—figure supplement 1*). Similar results could also be obtained via an alternative approach, in which instead of using noise components, principal components were shuffled across trials in a fashion that preserved their underlying correlation with chosen value, or by examining the contribution of PC1 or PC2 alone (*Figure 4—figure supplement 2*).

We then asked whether chosen value coding was reduced more by *local* (within-region) neural dynamics, compared to *global* (whole-brain) dynamics. We found a smaller but significant reduction could also still be found even if these 'local' principal components (i.e. from the same brain region) were orthogonalised with respect to those of another, simultaneously recorded brain region, when compared to performing the same analysis in reverse (*Figure 4—figure supplement 3*). This implies that some of the neuronal variance attributed to chosen value correlates originates as a consequence of the speed and amplitude at which dynamics unfold locally within a particular cortical area, and demonstrates the utility of our single-trial index for capturing these local dynamics.

## OFC and ACC dynamics have distinct influences on delay- and effort-based decisions respectively

ACC and OFC have established roles in effort- and delay-based decisions respectively (*Rudebeck et al., 2006*; *Prevost et al., 2010*; *Croxson et al., 2009*; *Kurniawan et al., 2013*; *Kable and Glimcher, 2007*; *Parvizi et al., 2013*). Because our LFP decomposition indexes decision dynamics locally, we hypothesised that the internal dynamics of ACC and OFC might preferentially influence activity in DLPFC on different trial types. Specifically, we predicted that firing rates in DLPFC would be preferentially affected by ACC internal dynamics on effort-based trials, but by OFC internal dynamics on delay-based trials. We could test this hypothesis by examining sessions in which DLPFC, ACC and OFC were all recorded simultaneously. We built a regression model in which LFP-derived PC2 weights from OFC and ACC competed for variance in explaining neural firing in DLPFC (see Methods). We estimated this separately on delay and effort trials. Both areas were found to influence DLPFC activity (*Figure 5—figure supplement 1*), but strikingly, ACC explained more variance in DLPFC firing on effort trials than delay trials (*Figure 5A*, magenta), whereas the converse was true for OFC (*Figure 5A*, black). This was not found to be true for analyses performed in the reverse direction (using DLPFC/ACC PC2 weights to explain OFC firing, or DLPFC/OFC PC2 weights to explain ACC firing).

Notably, OFC and ACC PC2 weights modulated DLPFC neuron firing from around 200 ms after choice onset (*Figure 5—figure supplement 1*), consistent with the time at which DLPFC neurons

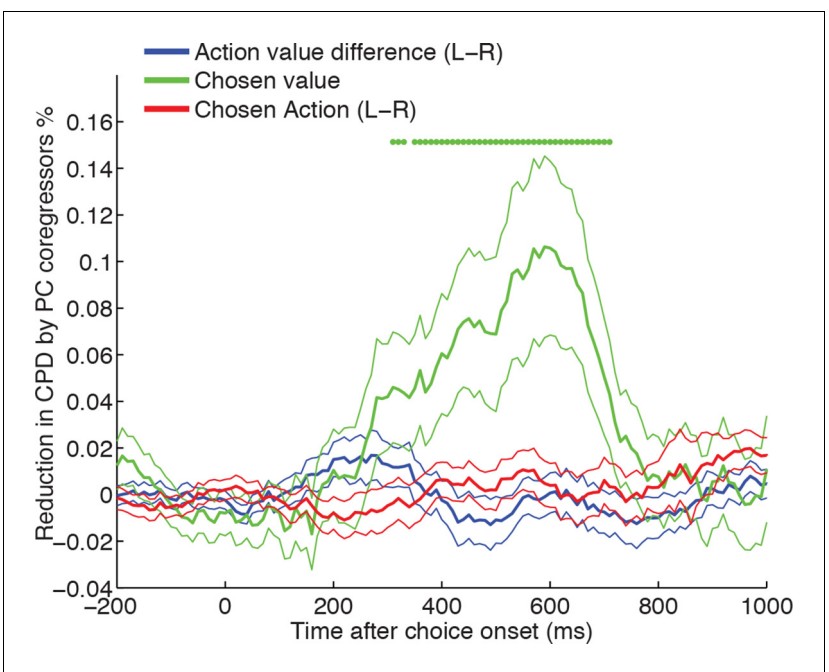

**Figure 4.** Chosen value, but not action value difference or chosen action, is explained away by neural dynamics. CPD for chosen value (green) is reduced by including PC1/PC2 from the LFP decomposition as coregressors in the decision model. The reduction in CPD (n=205 DLPFC neurons) for each of the decision variables in *Figure 1C* as a consequence of including PC1/2 is shown, by subtracting it from a control model that included two noise components (PC101/102). Note that the maximal value that this reduction can take is determined by the CPD for each regressor at each point in time (shown in *Figure 1B*). Lines show mean /- s.e. across neurons. Dots denote timepoints with a significant (p<0.05, permutation test) change in CPD across the DLPFC population. See also *Figure 4.* – figure supplement 1 for OFC/ACC, and *Figure 4—figure supplement 2* for the contribution of PC2 alone. *Figure 4—figure supplement 3* compares the effect of local DLPFC PC weights with respect to those from a distal, simultaneously recorded brain region (i.e. either OFC or ACC).

The following figure supplements are available for figure 4:

**Figure supplement 1.** CPD for chosen value is reduced by including PC1/PC2 as coregressors in the decision model across all three regions.

**Figure supplement 2.** CPD for chosen value is reduced by including PC1/PC2 alone as coregressors.

**Figure supplement 3.** Local PC1/2 (controlling for larger-scale global influences) reduces chosen value CPD more than global PC1/2 (controlling for local influences).

first begin to encode choice values (*Figure 1B*). These effects on DLPFC firing were maintained for several hundred milliseconds. Critically however, just prior to when response coding in DLPFC was peaking (around 800 ms, see *Figure 1B*), the contribution of ACC and OFC PC2 to DLPFC neuron firing changed in a cost-specific way. OFC PC2 began to selectively explain spiking variance on delay trials, whilst ACC began to explain spiking variance on effort trials (*Figure 5—figure supplement 1*).

We also found that when we performed a median split on DLPFC neurons by the degree to which they encoded chosen value in the analysis shown in *Figure 1B*, neurons with high chosen value selectivity were also those preferentially influenced by other regions' dynamics (*Figure 5B*). If DLPFC chosen value coding is interpreted as reflecting the speed at which a decision process unfolds, then these findings imply that the influence of one region's internal dynamics on another region's dynamics can be flexibly reshaped according to current task demands. This was not true, by contrast, when a median split was performed based upon the response selectivity of the DLPFC neuronal population.

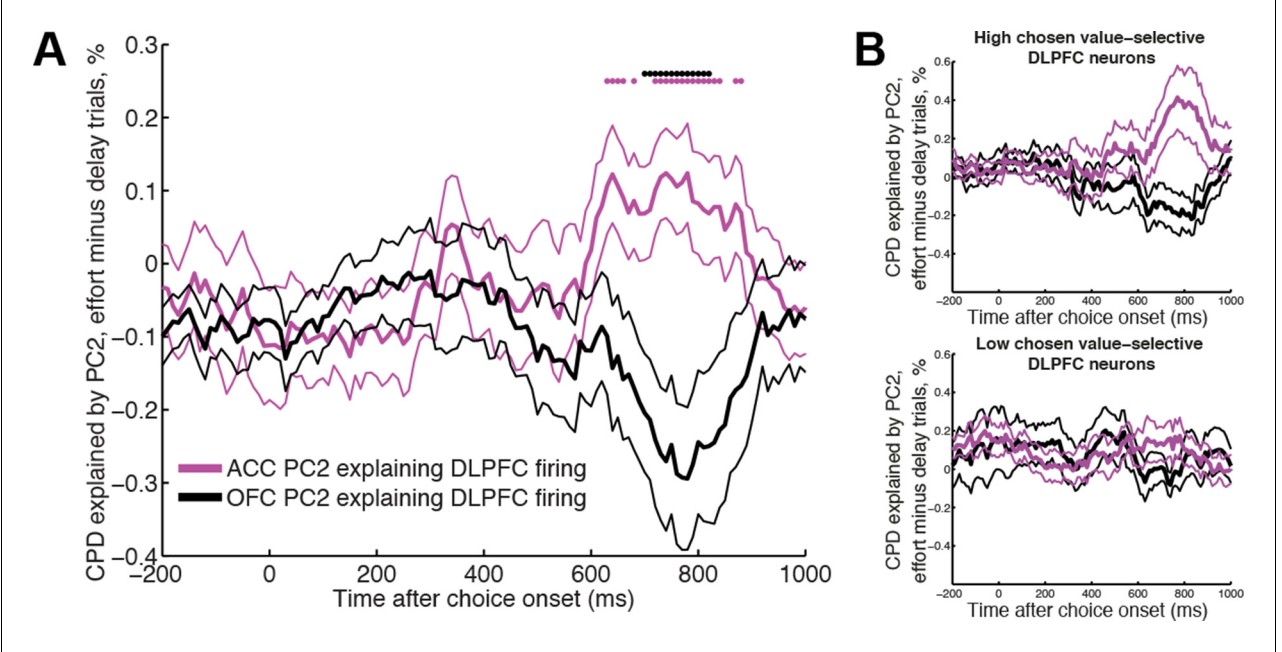

**Figure 5.** ACC and OFC local dynamics explain greater DLPFC neural firing on effort-based and delay-based decisions, respectively. (**A**) The relative CPD for DLPFC firing explained by PC2 on effort trials minus delay trials, where PC2 is derived from simultaneously recorded ACC LFP (magenta) and OFC LFP (black) (n=124 DLPFC units which had simultaneous recordings in both OFC and ACC). ACC explains more variance on effort than delay trials (positive-going values), whereas the converse is true for OFC (negative-going values). See also *Figure 5—figure supplement 1*. Lines show mean /- s.e. across neurons. Dots denote timepoints with a significant ($p<0.05$, permutation test) change in CPD across the DLPFC population. (**B**) The same analysis as in *Figure 5A*, having first performed a median split for chosen value-selectivity. Neurons with high chosen value selectivity (top panel, n=62 DLPFC units) show the effect found in *Figure 5A*, whereas neurons with low chosen value selectivity (bottom panel, n= 62 DLPFC units) do not.

The following figure supplement is available for figure 5:

**Figure supplement 1.** Cross-regional interactions, separately for delay versus effort trials.

## Similar single-trial choice dynamics can be obtained from human MEG data

The LFP value correlates in *Figure 1E* showed similar temporal profiles to a previous study in which human subjects made binary value-guided choices whilst undergoing MEG (*Hunt et al., 2012*). We therefore asked whether the MEG signal from this study, in the PFC subregion where this temporal profile had been observed (ventromedial prefrontal cortex, MNI 6, 28, –8 mm), could also be subjected to a similar single-trial PCA decomposition as our LFP data. Each subject provides a single 'virtual electrode' from which observations are made, and so data were stacked to form the matrix **X** with dimensions nSingleTrials ([=nTrials*nSubjects]) by nTimebins (see Methods). This was then decomposed as for the macaque data. We found a similar relationship between PC1 and PC2 (*Figure 6A*) controlling waveform amplitude and latency (*Figure 6B*). Moreover, PC2 had a similar relationship to chosen value and error trials as in macaques (*Figure 6C*). Finally, in this experiment, we also had a direct behavioural readout of decision formation on every trial as subjects could respond at any time after decision onset. Subject reaction times were strongly negatively predictive of PC2, over and above any contribution of chosen value, unchosen value or errors (*Figure 6C*). Our observations here provide a link between the dynamics of choice at the mesoscopic scale in macaques and its macroscopic counterpart in humans.

## Several features of the data are explained by competition via mutual inhibition

The transition in value correlates observed at the macroscopic scale in human MEG can be explained with reference to a class of recurrent neural network models displaying *competition via mutual*

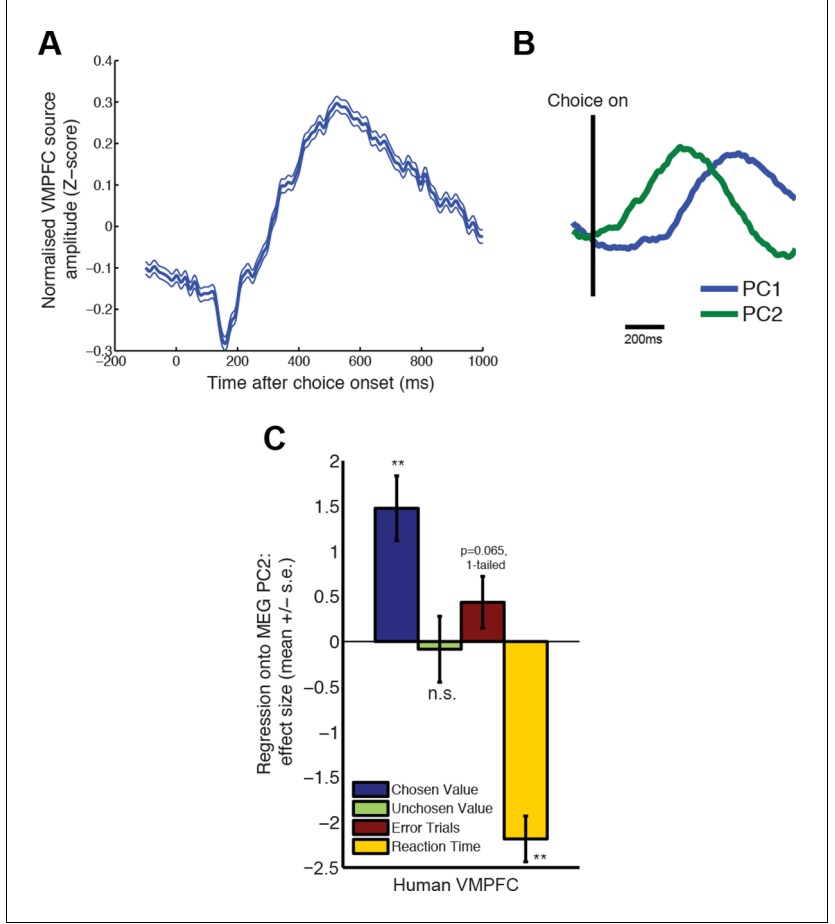

**Figure 6.** PCA decomposition of human MEG data shows similar characteristics to decomposition of monkey LFP data. (A) Averaged evoked response from data beamformed to ventromedial prefrontal cortex (MNI coordinate (6, 28, -8mm)), from a previous study of value-guided decision making (*Hunt et al., 2012*; *2013*). Lines show mean /- s.e. across trials. (B) PCA decomposition of human MEG data yields two principal components similar to those found in macaque LFP data (cf. *Figure 2C*). (C) The influence of chosen value, unchosen value and error trials on PC2 scores, estimated via multiple regression, in humans. Effects are similar to those found in macaque PFC (cf. *Figure 2E*). Also shown, in yellow, is the additional effect of reaction time (orthogonalised with respect to chosen value, unchosen value and error trials). Bars show mean /- s.e. across trials; ** denotes p<0.01, one-sample T-test.

inhibition (*Hunt et al., 2012*). Such models have success in explaining many features of neural data in perceptual choice (*Shadlen and Newsome, 2001*; *Wang, 2002*), focusing on how a decision might be implemented locally within a single cortical area of interest. We therefore asked whether these models might explain some of our observations at the microscopic (single-neuron) level, and whether a similar decomposition approach can be applied to mesoscopic (LFP) predictions from the network model as to our data.

We simulated a spiking attractor network model of choice, configured with the same connectivity structure and parameters as had been used in a previous study of perceptual decision-making (*Wang, 2002*) (*Figure 7A*; see also 'Supplementary details of network modelling'). We analysed the model predictions as we had the data, regressing action value difference, chosen value and chosen action onto single-neuron firing rates. Strikingly, value correlates in the network model possessed the same temporal profile as in DLPFC neurons (compare *Figure 1B* with *Figure 7B*). One discrepancy was the more sustained chosen value signal in the network model than in the data (which may result from inputs to the model persisting even after an attractor basin has been reached). We then ran a similar dimensionality reduction on summed network activity, a proxy for the model's LFP predictions, as we had done on both macaque LFP and human MEG data. By contrast with the LFP

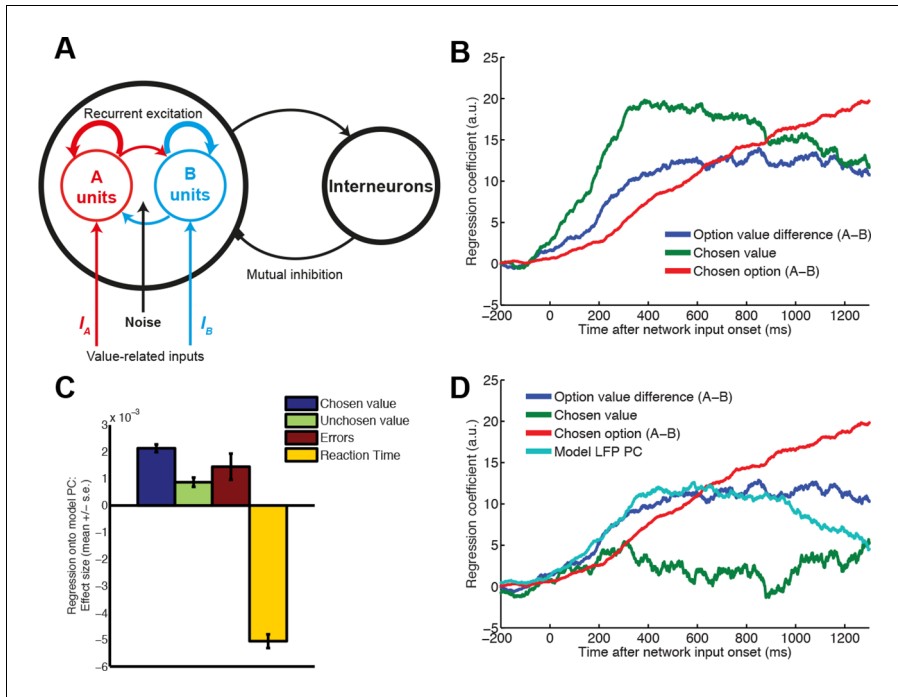

**Figure 7.** Relationship between value, LFP and single units explained by competition via mutual inhibition. (**A**) Model schematic. A and B units receive value-related inputs, integrate these via recurrent excitation, and competition via mutual inhibition. See 'Supplementary Details of Network Modelling' for details. (**B**) Correlates of decision variables in attractor network model single unit activity. Regression coefficient for option value difference (blue line), chosen value (green line) and chosen option (red line) onto firing rates of 'A' selective units in the network model. Compare with *Figure 1B*: in DLPFC, 'A' selective units are hypothesised to correspond to left selective units, and 'B' units to right selective units. (**C**) Regression of decision variables onto PC1 from PCA-decomposed LFP model predictions, as in figure 2e/f. Note that in model, PC1 captures variability in decision latencies, not PC2 as in data (see *Figure 7—figure supplement 1*). (**D**) Correlates of decision variables when model LFP PC1 is included as coregressor. LFP PC1 from model explains firing rates with a similar timecourse to LFP PC2 from data (*Figure 3C*), but explains away much of the contribution of chosen value to single unit firing (*Figure 4*).
The following figure supplement is available for figure 7:

**Figure supplement 1.** Dimensionality reduction of LFP predictions from attractor network model of competition via mutual inhibition.

data, we obtained a single principal component that controlled both waveform amplitude and latency (see 'Supplementary details of network modelling', and *Figure 7—figure supplement 1*). This suggests that within the network model there is covariation between these two features of the simulated ERP waveform, whereas in the data they are orthogonal. However, this principal component correlated with value in a similar fashion to both macaque and human data (compare *Figure 2E/6C* with *Figure 7C*). We then regressed the principal component back onto single unit firing rates, as in *Figure 3*. We found a similar timecourse of influence of the model's principal component on single unit firing to that found in the data (compare *Figure 3C* with *Figure 7D*, cyan). Moreover, we found that the internal dynamics of the model extracted from the PCA explained chosen value coding in a similar fashion to that observed in data (*Figure 4*), causing a marked reduction (compare *Figure 7B* with *Figure 7D*, green). Our observations in this study may therefore be explained mechanistically via a simple attractor network model of choice, which similarly captured LFP dynamics in our previous human MEG study (*Hunt et al., 2012*). Further details of network modelling are provided in Supplementary Information.

# Discussion

Many previous studies (*Kim et al., 2008*; *Daw et al., 2006*; *Padoa-Schioppa, 2013*; *Hampton et al., 2006*; *Padoa-Schioppa and Assad, 2006*), including our own (*Kennerley et al., 2011*), note the representation of chosen value in multiple brain structures during choice. Why this quantity is encoded so widely has been a matter of debate (*Rangel and Hare, 2010*). It is important to remember that this encoding comes from the viewpoint of the experimenter, seeking to describe variability across trials at a particular timepoint in the trial or decision process. As we have shown here, and would be predicted from previous behavioural studies (*Busemeyer and Townsend, 1993*; *Ratcliff and Rouder, 1998*; *Krajbich et al., 2010*), decision processes are dynamic and decision formation inherently exhibits variability across trials. This means that what appears to the experimenter as encoding of a decision related-signal may be a consequence of another ongoing process. To paraphrase a recent observation (*Cisek, 2006*), "the role of a decision-making system is to produce decisions, not to describe them."

Decision-making systems accumulate evidence in favour of different alternatives across time, and generate a categorical choice. This has been most carefully considered in integrate-to-bound models of perceptual choice and their neural correlates (*Gold and Shadlen, 2007*). Crucially, the time at which sensory evidence is maximally encoded in a neural integrator is not at the beginning of the decision (when no integration has occurred), nor at the end (when the bound has been reached on all trials), but in the *middle* of the decision process – when on some trials the bound has been reached, whereas on others little net evidence has been accumulated for each alternative. It can be very difficult in practice to infer when a decision begins or ends, and this poses difficulties when labelling a neural correlate as 'pre-decision' or 'post-decision'. The dynamical perspective instead explains certain correlates as emerging *as* the decision is formed. These signals may still remain of functional significance for other computations (such as subsequent computation of the reward prediction error [*Rangel and Hare, 2010*]), but our perspective on how they are generated is changed.

Value-guided choices are, like perceptual decisions, a dynamical process (*Busemeyer and Townsend, 1993*; *Krajbich et al., 2010*; *Summerfield and Tsetsos, 2012*). This brings into focus the potentially important role played by decision dynamics in generating correlates of chosen value. In the present study, we found that in DLPFC correlates of chosen value occurred maximally during the transformation from an initial representation of evidence (action value difference) to an eventual representation of choice (chosen action) (*Figure 1B*). A similar temporal progression could also be found in the network model of competition via mutual inhibition (*Figure 7B*). Crucially, however, a portion of the variance captured by chosen value was then explained away by including the speed and amplitude of the local LFP response on that trial as a coregressor, estimated via PCA decomposition of the LFP (*Figure 4*).

It is important to note that there are some caveats to this interpretation. Firstly, only some, not all, of the variance was removed by including PCA components as coregressors. As such, it may be the case that there is coding of chosen value during the decision that is not explained by the amplitude and speed of the evoked response during decision formation. However, despite the reasonable signal-to-noise ratio observed in LFP data, single-trial PC weights will likely contain a degree of observation noise. Hence our estimates likely form a lower bound on how much of the chosen value signal might be explained away by LFP dynamics. To address this question further, future studies might seek to estimate the degree of observation noise in estimating dynamics, and the theoretical limit that this imposes on how much variance could be explained in neural firing. A further important caveat is that chosen value correlates may be explained by different mechanisms at different points in the trial. LFPs are often interpreted as measurements of local synaptic input, and this may imply that our principal components mediate chosen value representations at the level of single units. It is also clear from *Figure 1—figure supplement 1* that correlates of chosen value are present in single unit firing until the end of the decision epoch. Future work may investigate the origin and functional significance of this persistent chosen value coding, perhaps for use at later task stages.

Our single-trial index also provides critical information about how dynamics simultaneously unfold at different speeds in different brain areas. This was evidenced most clearly by examining the interaction between OFC and ACC principal component weights, and DLPFC neuronal firing. Previous work has suggested a functional specialisation of OFC and ACC for delay- and effort-based decisions, respectively. In rats, lesions made to OFC lead to impulsive choices (of a small, immediate

reward over a larger, delayed reward) whereas lesions made to ACC lead to less effortful choices (of a small, easily obtained reward over a larger reward demanding more work) (*Rudebeck et al., 2006*). In humans, functional imaging activations yield a similar double dissociation between effortful and delay-based decisions (*Prevost et al., 2010*; *Croxson et al., 2009*; *Kurniawan et al., 2013*; *Kable and Glimcher, 2007*), and further evidence can be drawn from the effects of electrical stimulation of ACC (*Parvizi et al., 2013*). Our results extend these findings by suggesting that the rate at which dynamics unfold in OFC preferentially influences a decision signal in DLPFC on delay-based trials relative to effort-based trials, whereas the converse is true for ACC's influence on DLPFC (*Figure 5*). This supports the view that the effective influence of one brain region on another is modulated by the type of decision currently being made (*Hunt et al., 2014*; *Tauste Campo et al., 2015*). We note, however, that our analysis does not test the possibility is that a third (unobserved) variable could jointly affect both regions, doing so differentially on effort vs. delay trials.

Variation in ERP latency also corresponds to a shift in the phase of an oscillation (at low frequencies, present in the evoked response). Consistent with this, we found PC2 weights correlated with the phase of oscillations in the theta range (4-–8 Hz) as the decision was made (*Figure 2—figure supplement 3*). This suggests a possible way in which future studies may link our measure of LFP latency to spike-LFP coupling (*Koralek et al., 2013*; *Canolty et al., 2010*) or inter-regional LFP coherence (*Nacher et al., 2013*) during cognitive tasks.

The network model attempts to capture the dynamics of a single cortical region, but not interactions between regions. In our study, competitive dynamics similar to the model (saccadic action value difference transforming into chosen action) were clearly visible in DLPFC (*Figure 1B/7B*), but not in other regions (*Figure 1—figure supplement 1*). However, at the mesoscopic scale, dynamics of chosen value correlates were relatively indistinguishable across regions (*Figure 2E*, *Figure 1—figure supplement 3*). One explanation for these phenomena is that competition proceeds in a distributed fashion across multiple areas (*Rushworth et al., 2012*; *Hunt et al., 2014*; *Cisek, 2012*). Competition in DLPFC might occur selectively in saccadic action space, but this might be complemented by parallel competitions in other regions in other reference frames. Examples of such competitions might include those over particular decision attributes (e.g. OFC for delay, ACC for effort/physical action value), abstract goods (*Padoa-Schioppa, 2013*; *Padoa-Schioppa and Assad, 2006*), internal state variables (*Bouret and Richmond, 2010*), attentional allocation (*Lim et al., 2011*), goal or task-relevant prioritisation (*Hunt et al., 2014*; *Hare et al., 2009*) or other reference frames critical for the decision at hand (*Hunt et al., 2013*). Though lesion evidence clearly implicates areas like ACC, OFC and ventromedial PFC as critical and dissociable in value-based decision-making (*Kennerley et al., 2006*; *Rudebeck et al., 2008*; *Camille et al., 2011*; *Noonan et al., 2010*), our results indicate that identifying 'chosen value' signals is not sufficient to understand the functional dissociations of these areas in the decision-making process. Further studies are needed to fully understand the reference frame in which different decision areas contribute to decision-making (*Hunt et al., 2013*; *2014*; *Boorman et al., 2013*), with a careful consideration of the relationship between underlying choice dynamics and neuronal correlates of value. Future recurrent network models of choice involving hierarchical competitions across multiple areas (*Hunt et al., 2014*; *Chaudhuri et al., 2015*) might also capture this distributed approach, and seek to explain our observations concerning the selective effort- and delay-based interactions across areas (*Figure 5*). A distributed account would predict that in other experimental paradigms, single neurons would show similar competitive dynamics to our DLPFC neurons but in complementary frames of reference (*Rustichini and Padoa-Schioppa, 2015*; *Strait et al., 2014*).

PCA is one of many possible approaches to obtaining a useful set of temporal basis functions to describe variation in ERP waveforms, and it may be improved upon by future investigations. We selected it for its simplicity, and its known properties of returning a temporal derivative when capturing evoked responses of variable durations (*Friston et al., 1998*; *Mayhew et al., 2006*; *Woolrich et al., 2004*). Directly computing the ERP waveform and its temporal derivative would also be a valid approach to derive temporal basis functions. However, this approach may face problems if different components of the evoked response do not covary with each other. For example, the large, fast evoked response observed across all regions within 200 ms of stimulus onset (*Figure 1D*) was comparatively small in the first two principal components (*Figure 2B*). This implies that cross-trial variation in this early sensory-evoked component occurred largely orthogonal to cross-trial

variation in the later (putatively decision-related) component. This feature of the data is identified automatically using PCA.

The LFP decomposition returned by PCA shows a notable similarity to that observed in decompositions of trial-averaged waveforms from multi-electrode recordings in motor cortex during movement (*Churchland et al., 2012*). This provides links to dynamical systems perspectives on such activity. Unlike previous studies, however, our approach leverages mesoscopic dynamics containing high signal to noise ratio and reproducibility to extract single-trial information. This dispenses any requirement to first relate neural activity to experimental variables by averaging or regression (*Churchland et al., 2012*; *Mante et al., 2013*). We contend that this is particularly important in studying PFC and other areas distant from sensory input or motor output, whose relationship to experimental variables may often be considerably more complex than the experimenter envisaged (*Rigotti et al., 2013*).

## Materials and methods

### Neurophysiological procedures (monkey)

Full details of neurophysiological recording procedures are detailed in (*Hosokawa et al., 2013*) and (*Kennerley et al., 2009*), and precise recording locations in (*Hosokawa et al., 2013*). In brief, four male rhesus macaques served as subjects. Arrays of 10-–24 tungsten microelectrodes (FHC Instruments) were lowered acutely each day. Recordings were made from dorsolateral prefrontal cortex (DLPFC; primarily the dorsal bank of the principal sulcus), orbitofrontal cortex (OFC; primarily areas 11 and 13 between the medial and lateral orbital sulci) and anterior cingulate cortex (ACC; primarily area 24c in the dorsal bank of the cingulate sulcus). Recordings were also made from two subjects in the cingulate motor area, but this region is not considered in the present study. We excluded 5 sessions in monkey B that contained exclusively effort-based or delay-based decisions, 1 further session in monkey B where LFP data was not recorded, and 2 sessions in monkey E where the LFP data was heavily artifact-contaminated. Any electrode which did not contain a well-isolated neuron, and so might not lie within grey matter, was not included in the LFP analysis. From visual inspection of the evoked data, we further excluded from further analysis ~4% of individual electrodes that contained a high degree of artifact in the LFP recording. The numbers of recording electrodes included from each subject, after all exclusion criteria were applied, are shown in *Figure 1—figure supplement 2*. (As multiple units could occasionally be isolated from the same electrode, the total number of neurons exceeded the total number of electrodes for each region.) All procedures were in accord with the National Institute of Health guidelines and the recommendations of the University of California Berkeley Animal Care and Use Committee.

### Experimental task (monkey)

Full details of the experimental task are detailed in (*Hosokawa et al., 2013*). In brief, subjects were well-trained on the expected value of a set of 32 pictures, 16 of which predicted a quantity of reward (fruit juice) and associated effort to obtain reward, and 16 of which predicted a quantity of reward and an associated delay to reward (*Figure 1A*). The costs and benefits were titrated such that choice probabilities were approximately equally (and linearly) affected by both cost and benefit (*Hosokawa et al., 2013*). Subjects performed a cost-benefit decision task where they chose between two pictures on each trial. On half of the trials ('effort' trials) they chose between a pseudorandomly selected pair of the 'effort'-associated pictures. On half of the trials ('delay' trials) they chose between a pseudorandomly selected pair of the 'delay'-associated pictures. Effort and delay trials were interleaved.

On each trial, subjects fixated a central fixation point for 1 s, followed by the appearance of the two pictures on left and right sides of the screen for 1 s ('choice phase'). Throughout this choice phase, the monkey held fixation. Importantly, this meant that the time period when the monkey could respond (after the onset of the go cue, at 1000 ms) was outside of the window of our analyses, and so there was no potential behavioural (motoric) confound in neural analyses. This was with the exception of monkey K, who could not be sufficiently well-trained to fixate and so was free to saccade during the 1 s period. After the appearance of the go cue, the monkey saccaded to the preferred picture (except monkey K, who made a joystick response to the side with the preferred

picture). Subjects received juice reward after the 'cost' (effort/delay) was delivered. Only successfully completed trials are included in our analysis.

## MEG recordings and experimental task (human)

Full details of the experimental task, MEG data acquisition and analysis protocols are provided in (*Hunt et al., 2013*). Briefly, 18 subjects chose between two risky prospects consisting of differential levels of monetary reward magnitude and probability. Subjective values for each option were estimated using Prospect theory. We analysed data from 'comparison' trials (where both options appeared simultaneously, and subjects were free to respond at any time after decision onset). The analysed data were beamformed to a region of ventromedial prefrontal cortex (MNI coordinates = 6,28,−6 mm) previously found to contain dynamics analogous to those of the biophysical network model (*Hunt et al., 2012*). All subjects provided informed consent in accordance with local ethical guidelines.

## Regression analysis of macaque single neurons (*Figure 1B/C*, *Figure 1—figure supplement 1*)

Well-isolated single units (see [*Hosokawa et al., 2013*] for details) were timelocked to the onset of the choice epoch of successfully completed trials to create rasters (lasting from 1 s prior to choice onset to 2 s after choice onset). Each trial's data was then convolved with a boxcar to estimate the local average firing rate of the neuron on each trial in 200 ms sliding bins. These binned data were then regressed, across trials, against five variables using ordinary linear regression: a constant term for effort trials, a constant term for delay trials, the value difference between left and right options, whether the subject chose left or right, and the value of the chosen option. For each neuron, we calculated the coefficient of partial determination for each factor as in (*Kennerley et al., 2011*):

$$CPD(X_i) = [SSE(X_{-i}) - SSE(X)]/SSE(X_{-i})$$

where $SSE(X)$ refers to the sum of squared errors in a regression model that includes a set of regressors X, and $X_{-i}$ is a set of all the regressors included in the full model except $X_i$. In *Figure 1B*/ *Figure 1—figure supplement 1*, we plot the mean /- s.e. across all neurons recorded in a given brain area (for sessions that were also included in the LFP analysis). In *Figure 1C*, we calculated the Z-statistic for action value difference at 300 ms and chosen action at 700 ms for each neuron from the regression model, and plotted the relationship between these Z-statistics across all DLPFC neurons.

## Regression analysis of macaque LFP (*Figure 1*, *Figure 1—figure supplement 3*)

LFP data were downsampled from 1 KHz to 100 Hz, and timelocked from 500 ms before to 1000 ms after the onset of the choice phase. The average event-related waveform was computed for each electrode, and the grand mean /- s.e. of all electrodes within each brain region was plotted (*Figure 1D*). Based on previous behavioural modelling of the task(*Hosokawa et al., 2013*), value for each option was defined as reward level (scaled from 1 to 4) minus cost level (scaled from 1 to 4). For *Figure 1E*, we estimated the influence of chosen value and unchosen value on the evoked response at each timepoint using linear regression across trials, plotting the mean /- s.e. of the Z-scored regression coefficient across electrodes. In *Figure 1—figure supplement 3*, we split this regression into chosen and unchosen reward and cost. For *Figure 1F*, we estimated the local temporal derivative of the grand mean event-related potential, averaged over a local sliding window of 80 ms.

## Dimensionality reduction of LFP/MEG data via principal components analysis (*Figure 2*)

To extract single trial information from the local field potential (LFP), we used principal components analysis (PCA) of event-related data. Single-trial LFP data were timelocked from 200 ms before to 1000 ms after the onset of the choice phase. Separately for each subject, all data from all electrodes in OFC, ACC and DLPFC were stacked to form a large matrix **X**. The dimensions of **X** are [nRecordedElectrodes*nTrials] by nTimepoints. Each row of **X** corresponds to the LFP data collected from a single trial, on a single electrode.

To extract single trial information from the human data, a similar 'stacked' matrix was formed, but here single-trial data across all subjects were stacked, to make a matrix **X** with dimensions [nSubjects*nTrials] by nTimepoints. Each row of **X** corresponds to the beamformed data collected from a single trial, in a single subjects. (One difficulty faced by this approach is that beamforming contains a singular value decomposition (SVD) step to determine the orientation of sources, and when beamforming is run separately on each subject, then the meaning of a positive-going and negative-going deflection can be different across subjects because of an arbitrary sign flip induced by SVD. To overcome this, we used an iterated procedure that selected a sign for each subject that maximised the correlation coefficient in the evoked response, between subjects, and then multiplied each subject's data by the appropriate sign. Note that this sign-flipping correction does not produce changes in the PCA decomposition of the data, but instead simply ensures that the principal component weights can be interpreted in a consistent manner across subjects. It would not be a necessary step were a similar analysis to be run on a scalp potential, for example, whose sign is already consistent across subjects.)

The mean timecourse of the data was removed prior to running PCA, such that the principal components then captured *cross-trial variability* in the shape of the event-related waveform. We also removed trials that potentially contained non-neuronal (e.g. movement, electrical) artefacts. To achieve this, within-trial variability was indexed by taking the square root of the standard deviation of the event-related waveform across time. We excluded any trials whose within-trial variability index lay more than 2.32 standard deviations above the mean (>99th percentile).

In macaque data, most trials occur more than once in **X**, as in each recording session multiple LFP electrodes are recorded simultaneously and so there is a separate row for each electrode. Simultaneous multielectrode recording is not fundamental to the analysis approach of extracting single-trial dynamics. However, it becomes important when relating LFP and cellular data, below. In human data, each trial occurs only once, as there is one 'virtual electrode' per subject.

PCA was performed using the *svd* function in MATLAB. To ensure components had similar interpretations across subjects, we constrained PC1 to be positive 190 ms after stimulus onset, and PC2 to be negative 530 ms after stimulus onset. This constraint can be reasonably imposed by flipping the sign of both the principal component and its corresponding PC weights where necessary (as the sign of components returned by singular value decomposition is arbitrary). After this constraint was applied, highly similar components were observed across subjects (*Figure 2C*). In all subjects, PC1 controlled waveform amplitude, and PC2 controlled waveform latency. However, the timecourse of monkey K's principal components were notably different in latency from other subjects' waveforms (*Figure 2C*), and also had somewhat different value correlates, particularly in PC1, likely due to an inability to sustain fixation. For this reason, *Figure 2E* and *Figure 2—figure supplement 4* excludes data from monkey K, and monkey K's PCA value correlates are plotted separately in *Figure 2—figure supplement 3*. Importantly, this does not affect the interpretation of PC2, whose effects remain similar in monkey K as in other monkeys.

We note that it would have equally been possible to run dimensionality reduction on the data from a single electrode, in a single session. However, the difficulty faced with such an approach is how then to match components between different components and sessions, such that the components from the different PCAs have the same interpretation. The stacked PCA approach here is analogous to that used in 'dual regression' of functional MRI data, where resting state networks are matched across subjects by stacking all subjects into a large matrix before running dimensionality reduction (*Filippini et al., 2009*). It is appropriate due to the high degree of consistency of ERP waveforms (*Figure 1—figure supplement 2* and *Figure 2—figure supplements 1/2*).

## Regression analysis of single neuron data, using principal components derived from simultaneously recorded LFP (*Figures 3–5* and associated figure supplements)

Single neuron data were rasterised and binned as for the regression analysis of decision variables. For main *Figure 3C*, a regression model was estimated containing 19 coregressors to model out the effect of all experimental variables (see *Supplementary file 1* for a full list), plus the regressor of interest (trial-by-trial weights of PC2, derived from PCA decomposition of LFP from a simultaneously recorded electrode in another part of DLPFC). The CPD for the LFP-derived PC2 on single neuron firing was then estimated from this model. To calculate the percentage of individual significant

neurons (in main text), we estimated a significance criterion that controlled for multiple comparisons across time empirically from the data, by repeating the regression model but permuting the design matrix (*Nichols and Holmes, 2002*). Within a time window of 250 ms-–750 ms after choice onset, this yielded a p<0.01 significance criterion of Z>2.99 for significant positive responses, and Z<-3.05 for significant negative responses. Any neurons exceeding these thresholds at any point during this time window are reported as significant. We found the CPD for PC2 remained similar when a reduced model of experimental variables was used, containing 6 coregressors rather than 19 – a constant term for each trial type, an indicator variable for the chosen action (L-R), the action value difference between left and right options, and the chosen value. For *Figure 3A/B*, we used this model and plotted CPD for action value difference, LFP-derived PC2 and chosen action in two single-neuron examples. For *Figure 4* and associated figure supplements, we repeated this model, but subtracted the CPD for chosen value, action value difference and chosen action for a model that included PC1 and PC2 as coregressors, versus a model that included PC101 and PC102 (to account for any potential general reduction in CPD by including coregressors). Positive values on this figure therefore indicate a reduction in explained variance by experimental factors when PC1 and PC2 are included as coregressors in the model. Note that these analyses required simultaneously recorded LFP from another electrode in the same brain region, but this was only available for some recording sessions. The number of recorded neurons in each analysis was therefore smaller than in *Figure 1/ Figure 1—figure supplement 1* (DLPFC: n=205 neurons; OFC: n=114; ACC: n=194). For *Figure 4— figure supplement 3*, we sought to explore whether local LFP principal components provide greater contributions to the reduction in chosen value variance than those recorded from other areas. We therefore repeated the same analysis as in main *Figure 4*, but first orthogonalised local (DLPFC) PC1/2 weights with respect to PC1/2 weights from another simultaneously recorded area (either OFC/ACC). However, because both electrodes will contain observation noise on every trial, this analysis alone may not fully control for distal brain region effects. To provide a fairer comparison, we therefore performed the same analysis in reverse: examining the effect of the distal OFC/ACC PC1/ 2 weights, orthogonalised with respect to the local DLPFC PC1/2 weights. *Figure 4—figure supplement 3B* shows the latter analysis (distal orthogonalised with respect to local) subtracted from the former (local orthogonalised with respect to distal).

For *Figure 5* and associated figure supplement, we examined the 40 recorded sessions where DLPFC, OFC and ACC recordings were made simultaneously. Single neuron firing rates from the 124 DLPFC units were explained using a model that contained separate 6 terms: separate constant terms for effort and delay trials, separate LFP-derived PC2 weights from OFC for delay and effort trials, and separate LFP-derived PC2 weights from ACC for delay and effort trials. *Figure 5—figure supplement 1* shows the separate effects of PC2 from ACC and OFC for delay and effort trials respectively, on DLPFC firing; *Figure 5* shows the difference in CPD for effort minus delay trials for each region.

Significance for neuronal populations in *Figures 4,5* was assessed using a non-parametric permutation test (*Nichols and Holmes, 2002*). At each timepoint, for each neuron, we compared the CPD for the relevant variable of interest to the CPD for the same variable in a 500 ms pre-choice baseline period. Across the neuronal population, we tested whether this difference was significantly greater than zero, by comparing the mean difference to that of a null distribution, generated by randomly sign-flipping each neuron's effect and averaging across this permuted data. 10,000 permutations were used. We elected to use a non-parametric test as the difference in CPD will yield a non-Gaussian distribution; in practice, however, similar results could be obtained with parametric statistics (Student's T-test).

## Supplementary details of network modeling
## Methods
We adopted the same model as developed by (*Wang, 2002*) to make predictions of single neuron firing rates and regressions. The network consists of 2000 neurons, of which 400 are inhibitory interneurons and 1600 are excitatory pyramidal cells (see main *Figure 7A* for model schematic). The excitatory pyramidal cells fall into three categories: those selective for option A (240 cells), those selective for option B (240 cells), and the remaining, non-selective population (1120 cells).

## Structure of network connectivity

As in (*Wang, 2002*), the network possesses all-to-all connectivity but the strength of connections between cell types from different populations varies. The majority of connections have a synaptic strength $w$=1. Within the selective populations ('recurrent' connections from 'A' selective to 'A' selective and from 'B' selective to 'B' selective cells), however, excitatory connections are stronger, with a synaptic strength $w$=1.7. Between the selective populations (i.e. from 'A' selective to 'B' selective cells and vice versa), excitatory connections are weaker, with a synaptic strength $w_-$=0.8765 (= 1-$f$(w-1)/(1-$f$), where $f$=0.15 (fraction of cells in each excitatory pool). (Note that these weights are based on a Hebbian principle, as cells with similar selectivity are endowed with stronger connections, whereas those with different selectivity have weaker connections.) This network structure provides the network with its properties of integration via recurrent excitation, and competition via mutual inhibition (*Wang, 2002*).

## Selective inputs

Selective excitatory cells receive input, $I_A$ and $I_B$, whose firing rate is proportional to the value of option A or option B on each trial. For simulation purposes, the mean input rate, for each trial of both option A and B were independently drawn from a uniform distribution from 20 to 60 Hz, giving a fixed 'value' for each trial for each option, $\mu_A$ and $\mu_B$. Note that this differs from the model of (*Wang, 2002*) in that option A and option B are not anticorrelated with each other (appropriate to the task that we are modelling here). As in (*Wang, 2002*), at every 50 ms, the Poisson input rates to each population were resampled were independently resampled from Gaussian distributions with means $\mu_A$ and $\mu_B$, and standard deviation $\sigma$=10. Hence the two inputs vary stochastically in time (if $\sigma$ were to be set to 0, the two inputs would become constant across time).

## Background inputs; implementation of simulations

In addition to selective inputs, all neurons receive AMPA currents from background 'noise' inputs generated from a Poisson process with rate 2.4 kHz that varies independently from cell to cell. Both pyramidal cells and interneurons are described by leaky integrate-and-fire neurons using the exact same physiological parameters and dynamical equations as specified in (*Wang, 2002*). The network was initialised for 500 ms without selective inputs, and the decision was presented to the network for the subsequent 2500 ms (total simulation time of 3 s). The 'chosen option' was determined by the selective population whose average firing rate first exceeded 25 Hz, with the additional condition that the average firing rate of that population had to be 15 Hz greater than that of the other population. LFP predictions were generated by summing the firing rates of all cells (both selective and non-selective) in the network model. (Note that although LFP is thought to be more closely related to synaptic inputs than firing rates, input conductances and firing rates are essentially collinear within the model).

## Simulations and regression of single-neuron firing rates

We implemented the model in MATLAB and simulated 400 trials of data. The local firing rate of each population was estimated by applying a sliding window of 50 ms, and averaging across the firing rate of all cell types within a given population. To obtain the results in *Figure 7B*, we then regressed each population's firing rates against a model that contained a constant term, a term for whether the model chose A on each trial, the difference in value between option A and B (=$(\mu_A-\mu_B)$/80), and the value of chosen option (=$\mu_A$/80 on 'chose A' trials, $\mu_B$/80 on 'chose B' trials). We plot the regression coefficient of the 'A' selective cells as a function of time.

## Results

Main *Figure 7B* shows the structure of the network model. We first tried to replicate the transitions in value correlates that occur at the level of single neuron firing in DLPFC (*Figure 1B*). *Figure 7B* shows that by applying multiple regression, we reveal that in the selective populations of neurons, there is a variation across time in factors that influence their firing rates. Option values have a strong influence on firing rates of the selective neurons early in the trial (*Figure 4B*, blue), but later in the trial, as the network selects an option by approaching a stable attractor state, firing rates become better explained by the network's choice (*Figure 7B*, red). At the same time, the *chosen* value also

influences neuronal firing (*Figure 7B*, green). Note that this coding of chosen value happens across all neurons in the network (i.e. both selective and non-selective), whereas the transition from option value difference to chosen option only occurs in the selective cells.

Value correlates occurring at the level of the LFP across all three areas (*Figure 1E*; *Figure 1—figure supplement 3*) transition from initially reflecting value sum (chosen unchosen value) to later reflecting value difference (chosen-unchosen value). In a previous study, we showed that this transition is predicted to occur at the level of the LFP from the network model, and also that these predictions explained observations in human MEG data (*Hunt et al., 2012*). We therefore investigated whether we could apply a similar PCA decomposition to the model to that adopted to study our macaque LFP and human MEG data (main *Figure 2*), and also whether variability in LFP dynamics would similarly explain away chosen value coding in single neuron firing rates, as observed in *Figure 4*.

We ran a similar PCA decomposition on single-trial LFP predictions from the model to that run on data. The shape of the evoked LFP response from the model is shown in *Figure 7—figure supplement 1A*. Note that the evoked response from the model differs somewhat from the data. First, at the end of the trial, the model remains in a stable (working memory) attractor state with high activity, whereas the evoked response appears to return gradually towards its baseline level. This could easily be explained by the prepared action plan not being stored in prefrontal cortex, but in premotor regions. A second difference is that there is little to no variation in the single-trial *amplitude* of the response. Instead, the *principal* source of variability is in the reaction time, or rate of rise, of the network model. The only variation in amplitude is when the model reaches its final attractor state, but crucially this covaries with the speed at which the attractor basin is approached, and so with the rate of rise of the model. As such, the top two principal components (shown in *Figure 7—figure supplement 1B*) do not straightforwardly resemble the principal components in the data: it is *PC1* that captures the primary variation in response latency, playing a similar role to PC2 in the data (see *Figure 7—figure supplement 1C*). We therefore regressed single-trial model PC1 weights onto chosen and unchosen value, errors and reaction times, and found a similar pattern of results to that found in the data – a predominant influence of chosen value on single trial PC1 weights; a smaller, but positive influence of unchosen value on single trial PC1 weights, not found in the current dataset; a positive influence of error trials on network PC1 weights, and a strong negative influence of reaction times on network PC1 weights (*Figure 7C*).

Finally, we asked whether in the network model, chosen value variability is explained away by the internal dynamics of the model, as estimated by the LFP decomposition. We asked the same question of the model as we did of the data: does including PC1/2 from the model LFP decomposition cause a reduction in the regression for chosen value in model single neuron firing, but not for option value difference or for chosen option (*Figure 4*)? We found that it does (compare *Figure 7D* with *Figure 7B*).

## Acknowledgments

We thank H Barron, P Dayan, R Dolan, K Harris, M Rushworth, P Smittenaar and M Woolrich for comments on an earlier draft of this manuscript.

## Additional information

### Competing interests

TEJB: Senior editor, *eLife.* The other authors declare that no competing interests exist.

### Funding

| Funder | Grant reference number | Author |
| --- | --- | --- |
| Wellcome Trust | 098830/Z/12/Z | Laurence T Hunt |
| National Institute of Mental Health | R01-MH097990 | Jonathan D Wallis |
| National Institute on Drug Abuse | R21-DA035209 | Jonathan D Wallis |
| James S. McDonnell Foundation | JSMF220020372 | Timothy EJ Behrens |

| Wellcome Trust | WT104765MA | Timothy EJ Behrens |
|---|---|---|
| Wellcome Trust | 096689/Z/11/Z | Steven W Kennerley |
| National Institute of Mental Health | F32MH081521 | Steven W Kennerley |
| Wellcome Trust | WT088312MA | Timothy EJ Behrens |
| Wellcome Trust | WT080540MA | Laurence T Hunt |

The funders had no role in study design, data collection and interpretation, or the decision to submit the work for publication.

### Author contributions

LTH, SWK, Conception and design, Acquisition of data, Analysis and interpretation of data, Drafting or revising the article; TEJB, Conception and design, Analysis and interpretation of data, Drafting or revising the article; TH, Acquisition of data, Drafting or revising the article; JDW, Conception and design, Drafting or revising the article

### Author ORCIDs

Laurence T Hunt, http://orcid.org/0000-0002-8393-8533

### Ethics

Human subjects: All human subjects provided informed consent, including consent to publish. Ethical approval for this study was obtained from NHS Oxfordshire Research Ethics Committee C, approval reference 08/H0606/46.

Animal experimentation: Ethical approval was obtained for this study. All procedures were in accord with the National Institute of Health guidelines (Assurance Number A3084-01) and the recommendations of the U.C. Berkeley Animal Care and Use Committee (Protocol Number R283).

## Additional files

### Supplementary files

• Supplementary file 1. List of coregressors used when estimating contribution of LFP-derived PC2 to unit activity.

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
