## [Decision Letter]

[Editors’ note: a previous version of this study was rejected after peer review, but the authors submitted for reconsideration. The first decision letter after peer review is shown below.]

Thank you for choosing to send your work entitled "Capturing the Temporal Evolution of Choice in Prefrontal Cortex" for consideration at *eLife*. Your full submission has been evaluated by Eve Marder (Senior editor), Michael Frank (Reviewing editor), and two peer reviewers, and the decision was reached after discussions between the reviewers. Based on our discussions and the individual reviews below, we regret to inform you that your work will not be considered further for publication in *eLife* in its current form.

We should note first that all involved expressed interest in the findings and the high quality work, but questioned the significance and insight afforded by the work over and above what your group has published in the Nature Neuroscience article, which showed how the time course of various value-related signals are replicated using the attractor models. The main issues that came up in the review discussion (some of which are reiterated by the individual reviewer comments below) are as follows. First, both reviewers thought that the conclusions were overstated. They felt that while the PCA analysis was interesting, just because some of the variance in chosen value signals could be accounted for by a latency-based PC, that does not itself make the chosen value signal functionally irrelevant (epiphenomenal) (and even then, some variance remained). They also weren't compelled by the implication that the signals are pre-decisional rather than post-decisional, given that it is very difficult to infer when a decision was made (i.e. to show that the signal emerges when LFP is still ramping does not necessarily imply that it was prior to the decision). It was felt that it is equally plausible and perhaps more likely that CV signals are emerging *as* decisions are formed, and that CV signals have multiple functions, depending on the time point.

That said, we agree that the single-trial metric of LFP ERP latency does seem like a nice innovation. And its relationship to task variables (chosen value and error commission) provides some validation. But the reviewers were equally concerned about the relationship with effort vs delay signals given the timing of the observed variables, and the significance of the LFP latency metric's relationship to spiking activity seems murkier.

Thus we are rejecting the paper on the grounds that the main new analyses are not as compelling as would be needed for *eLife*. If you feel strongly that you can address these criticisms with new analyses and reframe the paper so that it is in-line with what can be concluded from the data given the concerns, we could consider a new submission, but again the new insights afforded beyond your previous work (beside the discovery of a PC component that relates to ERP latency) would have to be quite clear.

Reviewer #1:

This manuscript by Hunt et al. is based on the analyses of several different types of data collected from multiple experiments that have been described in previous papers. The most important piece of data was the single-neuron activity recorded from 3 different cortical areas, including the dorsolateral prefrontal cortex (DLPFC), anterior cingulate cortex (ACC), and orbitofrontal cortex (OFC). The authors have also analyzed the local field potential (LFP) data collected from these areas in the same experiment. The results obtained from these analyses were then compared to the results from the human MEG experiment and their network model, which is based on an influential model of Wang (2002). Through a set of clever analyses, the authors draw their main conclusion that a trial-by-trial index of decision dynamics, derived from the PCA analysis of LFP data, was largely correlated with the chosen value. Most interestingly, this index was used to demonstrate that the functional interactions between different cortical areas reflect the type of information considered during decision making (e.g., delay vs. effort). However, there are several major weaknesses in this manuscript.

1) The authors argue that the chosen value signals are epiphenomenon based on the fact that the chosen value signals encoded by individual neurons can be substantially diminished by the PCA-based trial-by-trial index of the decision process measured from the local activity. This argument is not convincing. It appears that the same logic, if applied to other types of signals encoded by any neurons, would make virtually any kind of neural signals epiphenomenon. In other words, the fact that a certain type of signal encoded by a given neuron can be accounted for by the local dynamics of its surrounding network does not mean that the encoded signal has no functional significance.

2) Although the dynamic changes in the functional connectivity identified by this analysis related to effort vs. delay condition are interesting, the time course of this difference is not consistent with the authors' conclusion. The results shown in Figure 5 indicate that the difference does not arise until 600 ms after the onset of the choice period. However, the chosen action signals in the DLPFC begin to emerge within 200 ms from the onset of the choice period, and reach its first asymptotic level before 400 ms. Therefore, the communication between DLPFC and ACC/OFC identified in this analysis is unlikely to be the main driver for the choice signals in the DLPFC.

3) It is not clear whether the use of PCA (i.e., PC2) is really necessary to derive the main conclusion of this study, because an alternative possibility is that the explanatory power of PC2 might be largely due to the similarity of its temporal profile with that of chosen value signals (shown in Figure 1). If this is the case, then the fact that PC2 is correlated with the chosen value signal (Figure 2) is not too surprising.

*Reviewer #2:*

This paper reports a monkey electrophysiology study of two-alternative value-based decision making, focusing on the temporal evolution of decision-related neural signals in DLPFC, ACC, and OFC. Main findings include the following:

a) DLPFC spiking activity appeared to progress from a graded encoding of the relative action values to a discrete encoding of the selected action.

b) LFPs in all 3 regions positively encoded the chosen option's value throughout the choice epoch, but shifted from a positive-signed to a negative-signed encoding of the unchosen option's value. This can be interpreted as a progression from encoding "value sum" to "value difference".

c) Single-trial LFPs were summarized using two PCA-based indices that reflected the overall amplitude and latency of the waveform, respectively. These indices scaled with the value of the chosen option, and also showed an association with spiking activity.

d) DLPFC spiking was associated with LFP latency in other regions. The association was stronger with OFC when delay-related costs were at stake, and stronger with ACC when effort-related costs were at stake.

Parallel analyses were applied both to a human MEG data set and to the behaviour of a neural network model, and some of the same effects were obtained. The application of directly parallel analyses to multiple modalities of neural data is a significant strength of the paper. The development of trial-wise indices of LFP latency is likewise a potentially important contribution.

Comments:

1) The paper has a very cogent Introduction pointing out the challenge of reading out static decision variables from dynamic processes. But some of the rhetoric around this topic seems to go beyond what the data support. The authors express skepticism that neural activity represents economic variables (the word "represent" frequently appears in scare-quotes). But the idea that neural signals evolve dynamically isn't incompatible with the idea that they represent economic variables. This paper's conclusions really seem to pertain to how, rather than whether, economic variables are neurally represented. The results suggest that chosen value, for example, is encoded in the LFP amplitude in a specific time window (e.g. around 400 ms where PC2 peaks, indicative of the latency of the underlying waveform), rather than, say, the overall peak of the evoked LFP.

2) Trial-by-trial LFP-derived indices are shown to be related to spiking activity, but it's not entirely clear how to relate this finding to what is already known about spike/LFP coupling. Could this effect be accounted for in terms of straightforward first-order correlations between the spiking and LFP time courses? What makes this seem plausible is that the time course of PC2 weights (Figure 2) looks similar to the time course of PC2 effects on spiking (Figure 3). Spikes and LFPs were recorded from different electrodes to avoid contamination, but it's still possible that a genuine relationship holds between the two (to mention just one previous example, a general negative correlation between multi-unit activity and LFP is shown in Whittingstall & Logothetis, 2009, Figure 1).

3) I don't follow the logic of the analysis in the subsection “Chosen value correlates as an epiphenomenon of varying decision dynamics”. In a regression model of spike rates, predictors for LFP-derived PC1 and PC2 steal variance away from a "chosen value" predictor but not from other task-related predictors. This is taken to imply that "chosen value" effects are mainly picking up on the latency of the neural response. But this result actually seems consistent with these effects being related to *either* amplitude (PC1) or latency (PC2). The latency interpretation would be on stronger footing if it could be shown that chosen value effects varied with the addition/removal of PC2 only.

4) I'm concerned about whether it's appropriate to treat LFPs from simultaneously recorded electrodes as independent observations. Panel A of Figure 4—figure supplement 2 shows that LFP features are highly correlated across electrodes, even in different brain regions. This probably isn't a problem for the PCA decomposition, but it does seem like an issue for inferential tests of associations between LFP and task variables. For example, Figure 1 plot "mean /- s.e. across electrodes"-but since multiple electrodes were recorded per trial, the nominal number of electrodes may be larger than the true number of independent observations. The same concern applies to the results in Figure 2 and related figures. It would be helpful for the paper to clarify how interdependence among electrodes was dealt with.

[Editors’ note: what now follows is the decision letter after the authors submitted for further consideration.]

Thank you for resubmitting your work entitled "Capturing the temporal evolution of choice in prefrontal cortex" for further consideration at *eLife*. Your revised article has been favorably evaluated by Eve Marder (Senior editor), Michael Frank (Reviewing editor), and three reviewers, including a new referee who was not involved in the previous round of review. They all agree that the manuscript has been improved and the novel contributions are clear and worthwhile relative to the previous work. However, there are some remaining issues that need to be addressed, as outlined below.

1) The authors claim that the PC2 from LFP provides a measure of local dynamics that is more specific (and relevant to unit activity) than larger scale global dynamics. The analysis provided by the authors (effect of local LFP after orthogonalizing on LFP from another region) is encouraging in this regard, however supporting the claim would also require showing that the opposite is not true (i.e. that running the analysis in reverse does not yield the same result). The correlation coefficients in the histograms of Figure 4—figure supplement 3 suggest that this may be the case, although there does not seem to be much of a within region advantage for DLPFC in particular. Also, as far as I can tell, the authors did not incorporate reaction time, which is typically used as a trial-by-trial indicator of decision dynamics, into these analyses. I suspect that this is because the required delay is sufficiently long to create a floor effect on RTs, however if this is the case it deserves mention in the text. Otherwise the authors should test whether LFP measures relate to the "chosen value" aspects of neural activity after accounting for RT.

2) The authors show that PC2 captures variance in neural firing that is typically ascribed to chosen value. I agree with the authors that this is an important point. Yet I found the interpretation of this point to be somewhat overly specific: "We therefore questioned whether chosen value coding might simply be a consequence of the same neural dynamics occurring at varying latencies across trials." This is certainly one interpretation of the authors' results, but not the only one. Another (more standard) interpretation of this finding is that the LFP components mediate the chosen value representations at the level of single units. Given that the LFPs are often interpreted as measurements of local synaptic input, I find this interpretation to be fairly reasonable and consistent with the data provided. The finding that PC1 (which does not contain timing information) also steals variance from chosen value may even provide some specific evidence for this interpretation. As I understand it, the main reason that the authors' interpret the data in terms of dynamics is provided in the final analysis showing that in an attractor network of decision-making the same sorts of signals emerge as a byproduct of competition. I think in order to improve the clarity it would be useful to avoid specific interpretations of causality until the Discussion section where both possibilities should be discussed.

3) Relatedly, another reviewer noted that the most direct evidence for the notion that chosen-value effects emerge from "the same choice dynamics occurring at different rates on different trials" is derived from the analysis on p. 10, but I'm afraid I'm still not onboard with the logic of this. The authors foreground an analysis showing that the chosen-value effect in spike rates is reduced by jointly including PC1 (indexing LFP amplitude) and PC2 (indexing LFP latency) in the regression model (Figure 4, Figure 4—figure supplement 1, and Figure 4—figure supplement 3). I still don't see how this is germane to their conclusion about latency. The specific effect of the latency-related PC2 is now shown as a supplement, and is considerably weaker albeit nonzero. The original chosen-value effect, which peaked at a% CPD of around 0.9 (Figure 1), is reduced by up to about 0.1 by both PCs together (Figure 4), or 0.03 by PC2 alone (Figure 4—figure supplement 2). So wouldn't it be at least as accurate to conclude that chosen-value effects emerge from the same neural dynamics occurring at different amplitudes across trials? (The predictions of the attractor network are also unclear in this regard, i.e. in the predictions for combination of PCs, and so it seems important that the authors clarify the predictions of the attractor model.) The same concern applies to the local-versus-distal result in Figure 4—figure supplement 3, which is interpreted solely in terms of neural latency but results for PC2 alone are not shown.

It seems to me an easier conclusion to square with the data would be that chosen-value effects originate from a combination of the amplitude and speed of the dynamically unfolding neural response. This may not be so starkly different from existing perspectives, although it's of course still valuable to see the details of how it plays out. The authors could also easily reframe the result that LFP measures partially mediate the chosen value effect and tone down the oppositional framing of decision dynamics versus static chosen value representation. They could use the network model results as a podium to say that chosen value representations need not emerge for the purpose of encoding chosen value, which would be entirely accurate from my standpoint.

4) A similar issue comes up in the section on neural network modeling, which shows that chosen-value effects in the simulation can be explained away by a principal component capturing variability in neural dynamics, but doesn't state clearly enough that "dynamics" here refers, not just to latency, but to a mixture of amplitude and latency (Figure 7—figure supplement 1, panel C). (Related to this, a sentence the sentence “we obtained principal components that controlled waveform amplitude and latency, as in the ERPs" seems inaccurate, since what was actually extracted was a single component that mixed amplitude and latency, unlike the earlier analysis.)

5) The authors use the first principal component of the attractor network model decomposition rather than PC2. While I agree with the authors that this component seems to contain some temporal information, it does seem to mainly capture amplitude. Given that the authors interpret PC2 as if it were the derivative of the ERP, it seems that it would be a bit easier to make the connection between computational model and biology if model signals were decomposed using separate components to capture overall amplitude and derivative. This could be done simply by creating weight vectors based on the average model response (ERP) and the derivative of that signal.

6) The authors note that PC1 relates primarily to value sum, however this does not seem to be the case in ACC, where PC1 takes positive coefficients for chosen value and negative coefficients for unchosen value. This makes me wonder whether the added contributions of PC1 in explaining single unit chosen value effects relate primarily to the inclusion of PC1 from ACC electrodes. In my opinion, such a finding would not necessarily detract from the dynamics idea, as overall ACC signal amplitude may play a role in adjusting decision threshold, which directly affects dynamics. Either way, the authors should make the heterogeneity across recording locations clear in this regard.

7) The section on inter-region correlations (“OFC and ACC dynamics have distinct influences on delay- and effort-based decisions respectively”) is quite specific about causal directionality, concluding that DLPFC activity is influenced by OFC and ACC. It isn't apparent to me that there's any support for this. (A third-variable explanation would be a plausible alternative.)

---

## [Author Response]

[Editors’ note: the author responses to the first round of peer review follow.]

*We should note first that all involved expressed interest in the findings and the high quality work, but questioned the significance and insight afforded by the work over and above what your group has published in the Nature Neuroscience article, which showed how the time course of various value-related signals are replicated using the attractor models.*

*[…] We could consider a new submission, but again the new insights afforded beyond your previous work (beside the discovery of a PC component that relates to ERP latency) would have to be quite clear.*

We thank both reviewers and the Reviewing editor for their careful reading of the paper and their enthusiasm for certain aspects of our results. Below, we address each of the reviewers’ comments point-by-point. First, however, we wish to briefly address the key insights and points of significance afforded by the present study that go beyond our previous work.

The main contribution of the paper is to explain the origins of some of the most commonly found signals during value‐guided choice. Uniquely to this study, we have attempted to do so in both single-unit physiology and human neuroimaging data.

Our paper provides two perspectives on value correlates found in these data. First, we establish the relationship between mesoscopic PCA-derived latency components and correlates of value in single-neuron firing, exploiting the single-trial nature of the measurements obtained from the PCA. Second, we test predictions of different value-related signals via the attractor network model in both forms of data.

Although different in approach, these two perspectives remain complementary to one another, as they both explain neural correlates of value in terms of variation in neural dynamics across trials. In both cases, we believe there to be substantial novelty and significance over and above what we have published previously.

With regards the single-trial PCA approach, both reviewers appreciated that this was a useful and potentially important innovation. The key advantage of our approach is that it provides single-trial estimates of the speed of decision formation from a neural measurement in multiple simultaneously recorded areas. As Reviewer #2 noted, it is a significant strength that this allows the same analyses to be performed on both human and monkey data, and that in the monkey data, the measure can then be related to spiking activity. As Reviewer #1 noted, the application of this approach to study functional interactions between OFC, ACC and DLPFC is an interesting and significant novel result for our understanding of cost-benefit decision formation. In short, this PCA approach provides a reliable internal measure of the speed at which a decision is made which could be useful for understanding the neural processes that regulate the speed of decision or action in a broad array of behavioural neuroscience research.

Whilst the novelty of this aspect of the paper appears not to be in question, both reviewers felt we had overstated some of the claims made. We agree with the reviewers on this point. We have therefore reframed the paper in line with their comments. Moreover, both reviewers had technical concerns about some of the analyses shown, and asked us to rule out alternative explanations of the PCA-derived effects. We have sought to address each of their concerns with several additional new analyses, which we detail in the point-by-point response to their reviews below.

Regarding the attractor network modeling of LFP and single-unit data, one of the overarching claims in both the present paper and our Nature Neuroscience (2012) paper is that the attractor network model of Wang (2002) is a useful framework for explaining the temporal evolution of signals during value-guided choice. A transformation seen in our previous paper (from signaling overall value to value difference in human MEG) is also seen in the LFP data in our current paper across all three brain areas. There are, therefore, some similarities between the ideas motivating both papers.

However, there are several novel findings in the present paper that make a substantial contribution beyond that of our previous paper. In particular:

a) In DLPFC, predictions of the model are borne out at the single‐unit level as well as at the mesoscopic scale of LFP signals (compare Figure 1 and Figure 7).

b) In DLPFC, the transformation from signaling action value difference to signaling the chosen action happens within the same neurons (Figure 1), matching the model’s prediction that pools of neurons selective for different options compete across time to become the selected option. Such a finding is consistent with value comparison being realized within DLPFC in an action frame of reference.

c) Chosen value signals are ubiquitous throughout many regions with similar latencies in both single-unit and LFP data, whereas only DLPFC shows action value difference to chosen action transformations (Figure 1—figure supplement 1). Such a finding is consistent with a model of distributed competition via mutual inhibition, as outlined in the discussion.

d) LFP value signals emerge at the time when the derivative of the LFP signal is non-zero (Figure 1), implying that value affects the rate of change of the mesoscopic signal, not the amplitude of the evoked response.

In short, we cannot accept that the present findings do not add additional significance and insight over what was in our previous Nature Neuroscience article. There is a substantial qualitative difference in the evidence being provided in the two papers.

Importantly, however, in the revised manuscript we have provided further additional analyses in support of our conclusions, which we hope address the reviewers’ concerns. These are outlined in the point-by-point responses below.

*[…] The main issues that came up in the review discussion (some of which are reiterated by the individual reviewer comments below) are as follows. First, both reviewers thought that the conclusions were overstated. They felt that while the PCA analysis was interesting, just because some of the variance in chosen value signals could be accounted for by a latency-based PC, that does not itself make the chosen value signal functionally irrelevant (epiphenomenal) (and even then, some variance remained). They also weren't compelled by the implication that the signals are pre-decisional rather than post-decisional, given that it is very difficult to infer when a decision was made (i.e. to show that the signal emerges when LFP is still ramping does not necessarily imply that it was prior to the decision). It was felt that it is equally plausible and perhaps more likely that CV signals are emerging* as *decisions are formed, and that CV signals have multiple functions, depending on the time point.*

We agree with both reviewers that some of the conclusions were overstated, and that we made the arguments of the paper too rhetorical. We have therefore reframed the tone of the paper throughout. In particular:

i) We strongly agree that the key idea is that chosen value signals are emerging ‘as’ a decision is formed. We did not intend to imply that the signals are pre-decisional rather than post-decisional. To clarify this, we have added to the Introduction:

“[…] the dynamical perspective proposes that they may in fact originate as a consequence of time evolving decision processes. Rather than casting a chosen value representation as signaling ‘pre-decision’ or ‘post-decision’ variables to downstream brain areas, it argues that such correlates inevitably emerge as a decision is made.”

To the Discussion, we have added:

“It can be very difficult in practice to infer when a decision begins or ends, and this poses difficulties when labelling a neural correlate as ‘pre-decision’ or ‘post-decision’. The dynamical perspective instead explains certain correlates as emerging as the decision is formed.”

ii) We agree with Reviewer #1 that there may still be functional relevance to chosen value signals, and that they may play different roles at different points in the trial. To make this more explicit in the Discussion, we have added:

“[…] These signals may still remain of functional significance for other computations (such as subsequent computation of the reward prediction error (Rangel and Hare, 2010).”

And:

“[…] A further important caveat is that chosen value correlates may be explained by different mechanisms at different points in the trial.”

iii) As Reviewer #2 correctly surmised, the paper’s conclusions really pertain to how certain value signals might originate, rather than whether they must then be functionally relevant or not (as implied by the term ‘epiphenomenal’).

We have therefore removed the claim that because the chosen value signal is reduced by the PCA-derived estimates, it is epiphenomenal. To give examples of the kinds of changes that have been made (there are many such changes throughout the paper):

“This implies that chosen value correlates are epiphenomena of the speed at which dynamics unfold locally within a particular cortical area […].”

has been replaced by:

“This implies that some of the neuronal variance attributed to chosen value correlates can be explained as a consequence of the speed at which dynamics unfold locally within a particular cortical area […].”

iv) We also note in the Discussion that it is important to remember that only some of the chosen value variance is explained using the single- trial PCA approach:

“It is important to note that only a portion, not all, of the variance was removed by this step. As such, it may be the case that there is coding of chosen value during the decision that is not explained the dynamics of decision formation.”

v) We have removed the usage of quotation marks around the word represent, which concerned Reviewer #2.

We hope that by reframing the paper in this way, we have made its tone less rhetorical and also made clearer what can and cannot be concluded from the data.

*That said, we agree that the single-trial metric of LFP ERP latency does seem like a nice innovation. And its relationship to task variables (chosen value and error commission) provides some validation. But the reviewers were equally concerned about the relationship with effort vs delay signals given the timing of the observed variables, and the significance of the LFP latency metric's relationship to spiking activity seems murkier.*

Reviewer #1 noted the potential interest of the PCA-based analyses between brain regions for effort vs. delay trials, but had concerns about the late timing of these effects. However, these were addressable concerns, and we have provided a response to them in the response to Reviewer #1. Reviewer #2 was concerned that the relationship between the LFP PCA and spiking could be more clearly related to existing knowledge about spike-LFP relationships. We have now addressed this point below.

Reviewer #1:

This manuscript by Hunt et al. is based on the analyses of several different types of data collected from multiple experiments that have been described in previous papers. The most important piece of data was the single-neuron activity recorded from 3 different cortical areas, including the dorsolateral prefrontal cortex (DLPFC), anterior cingulate cortex (ACC), and orbitofrontal cortex (OFC). The authors have also analyzed the local field potential (LFP) data collected from these areas in the same experiment. The results obtained from these analyses were then compared to the results from the human MEG experiment and their network model, which is based on an influential model of Wang (2002). Through a set of clever analyses, the authors draw their main conclusion that a trial-by-trial index of decision dynamics, derived from the PCA analysis of LFP data, was largely correlated with the chosen value. Most interestingly, this index was used to demonstrate that the functional interactions between different cortical areas reflect the type of information considered during decision making (e.g., delay vs. effort).

We thank the reviewer for these comments. It is clear that he/she understood well the close relationship between our current study, our previous MEG study (Hunt et al., 2012) and the Wang (2002) model. In considering the relationship between these different studies, we would like to reiterate the four key findings outlined above concerning the novelty of the present results over and above our previous findings (see point 2 in response to cover letter, above). We are unaware of any study that has used this model to explain the dynamics of single unit data in DLPFC, shown that the action value to chosen action transformation occurs within the same neurons, shown that chosen value single neuron correlates are found with similar strengths and time-courses across multiple simultaneously recorded brain regions, and shown that the LFP value signals emerge as the signal is ramping (when the derivative of the signal is high). We believe these empirical findings to be of considerable importance when evaluating the hypothesis that the Wang (2002) model explains single-neuron dynamics during value‐guided choice.

We are also pleased that the reviewer considered the PCA-based analyses of the LFP data to be both novel and interesting. However, it is clear that he or she felt we overstated our interpretation of the reduction in chosen value signals by the PCA-based single-trial index, and had two other technical concerns about our findings using this approach. As we outline below, we agree with the reviewer’s point 1 and have tempered our conclusions in line with his/her argument. We have also sought to address his/her points 2 and 3 below.

1) The authors argue that the chosen value signals are epiphenomenon based on the fact that the chosen value signals encoded by individual neurons can be substantially diminished by the PCA-based trial-by-trial index of the decision process measured from the local activity. This argument is not convincing. It appears that the same logic, if applied to other types of signals encoded by any neurons, would make virtually any kind of neural signals epiphenomenon. In other words, the fact that a certain type of signal encoded by a given neuron can be accounted for by the local dynamics of its surrounding network does not mean that the encoded signal has no functional significance.

Both reviewers felt that we had overstated the claim that could be made with our results, and we agree with Reviewer #1 that there are difficulties with labeling the chosen value signal as ‘epiphenomenal’. We have sought to address both Reviewer #1 and Reviewer #2’s concerns by changing the tone and conclusions of the paper throughout.

However, we disagree with the reviewer that the same logic would apply to signals encoded by any neurons. The PCA analysis causes a reduction in variance explained by chosen value, but the same reduction in variance is not found for either action value coding or chosen action coding, as is shown in Figure 4. This demonstrates that it is selectively chosen value coding which can be explained as a consequence of the speed at which dynamics unfold locally within a given area.

We refer the reviewers to our responses i) – v) above.

*2) Although the dynamic changes in the functional connectivity identified by this analysis related to effort vs. delay condition are interesting, the time course of this difference is not consistent with the authors' conclusion. The results shown in Figure 5 indicate that the difference does not arise until 600 ms after the onset of the choice period. However, the chosen action signals in the DLPFC begin to emerge within 200 ms from the onset of the choice period, and reach its first asymptotic level before 400 ms. Therefore, the communication between DLPFC and ACC/OFC identified in this analysis is unlikely to be the main driver for the choice signals in the DLPFC.*

We agree with the reviewer that our original interpretation of this result was problematic. Based on subsequent analysis of the data, we have removed our original interpretation of this finding, and rewritten this section in a way that more reasonably reflects the timings of the different effects shown.

In summary, there is good evidence that OFC and ACC both influence DLPFC at the times relevant for the decision process. However, we now remain more agnostic about why the difference between trial types emerges late in the decision phase than in our initial submission. This late timing is robust, replicating across two different groups of neurons. But in a further analysis, we find that it is particularly strong in DLPFC neurons that correlate with chosen value. We now offer one potential explanation for the late timing of the effect in the Discussion.

Firstly, we think it is important to consider Figure 5—figure supplement 1. This shows that although the difference between effort and delay trials for ACC and OFC only emerges late in the decision process, both regions have a significant influence on DLPFC firing far earlier in the trial.

We have now made this point more explicit in the revised manuscript:

“[…] Notably, OFC and ACC PC2 weights both modulated DLPFC neuron firing from around 200 ms after choice onset on both trial types (Figure 5—figure supplement 1), consistent with the time at which DLPFC neurons first begin to encode choice values (Figure 1). These effects on DLPFC firing were maintained for several hundred milliseconds.”

However, we previously concentrated on the late timing of the *difference* effect in Figure 5 (subtracting one trial type from another):

“[…] towards the end of the decision period, the contribution of ACC and OFC PC2 to DLPFC neuron firing changed in a cost-specific way. By subtracting one trial type from the other, this revealed that ACC explained more variance in DLPFC firing on effort trials than delay trials (Figure 5, magenta), whereas the converse was true for OFC (Figure 5, black).”

As the reviewer correctly surmised, our previous interpretation was that this reflected other regions’ influence on response-selective neurons, as this coincided with the largest CPD for chosen response in Figure 1. However, the reviewer points out that this effect first appears earlier in the trial. We therefore sought to test this hypothesis more explicitly. We repeated the ‘effort minus delay trial’ difference analysis, having first performed a median split of DLPFC neurons into those with high and low response-selectivity (Figure 8).

Author response image 1.Same analysis as in Figure 5, having first performed a median split for response-selectivity.Both neurons with high response selectivity (left panel, n=62 DLPFC units) and neurons with low chosen value selectivity (right panel, n= 62 DLPFC units) show the effect found in Figure 5.**DOI:**
http://dx.doi.org/10.7554/eLife.11945.024

The effect can be seen in both sets of neurons (including its late timing), suggesting it is a robust finding. However, it is not noticeably stronger in the response-selective neurons, contradicting our original hypothesis. We have therefore removed the about it being linked to the emergence of the response from the paper. We thank the reviewer for provoking this reanalysis of the data.

Instead, we considered whether the effect might be particularly prominent in other specific classes of neuron. We found that when we instead performed a median split of DLPFC cells by *chosen value* encoding, there was a clear difference in the degree to which neurons showed the between-region effect in Figure 5.

In light of these additional results, we have added this median split by chosen value as a separate panel to Figure 5:

“We also found that when we performed a median split on DLPFC neurons by the degree to which they encoded chosen value in the analysis shown in Figure 1, neurons with high chosen value selectivity were also those preferentially influenced by other regions’ dynamics (Figure 5). This was not true, by contrast, when a median split was performed based upon the response selectivity of the DLPFC neuronal population.”

Although this effect is maximal between 600 and 800 ms, there is evidence for it at earlier time-points in the trial (between 400 and 600 ms). This still leaves some ambiguity as to exactly why the effect is strongest towards the end of the decision period (Figure 5), and why before this time OFC and ACC both have an influence on DLPFC firing (Figure 5—figure supplement 1).

One possible explanation is that if local PC2 weights reflect the speed of OFC and ACC attractor dynamics, then it is only once these regions reach attractor basins that they begin to selectively influence neural activity in DLPFC. There may also be a delay in the time-course of this interaction between regions. We now put this forward as a potential hypothesis in the Discussion, but we make clear that it is a hypothesis requiring further investigation (please see: “Further, future spiking network models of choice involving hierarchical competitions across multiple areas […] the time-course of these inter-regional effects more explicitly.”

*3) It is not clear whether the use of PCA (i.e., PC2) is really necessary to derive the main conclusion of this study, because an alternative possibility is that the explanatory power of PC2 might be largely due to the similarity of its temporal profile with that of chosen value signals (shown in Figure 1). If this is the case, then the fact that PC2 is correlated with the chosen value signal (Figure 2) is not too surprising.*

We disagree with the reviewer’s comment here. The key point about the obtained principal component is that addition or subtraction of a single-trial estimate of PC2 weights tells us about trial-by-trial variation in dynamics. This is understood most clearly in the right-hand panel of Figure 2, which shows that adding or subtracting PC2 captures variation in the waveform latency. It therefore provides an index of decision speed. The clearest evidence of this is the strong negative correlation with reaction time in the human MEG data, where subjects were free to respond at any point in the trial (Figure 6). Crucially, the measure provides a local measure of decision formation that otherwise we would not have access to, and it does so on a single-trial basis. It is only because the PCA provides access to these single-trial estimates that we are able to deduce that the chosen value signal is related to internal decision speeds. But more importantly, these single-trial estimates are critical for allowing for subsequent analysis of how this relates to other measures, as can performed in the between-region analysis (Figure 5) or the coupling to ongoing phase of an oscillation (Figure 2—figure supplement 3, see response to Reviewer #2 question 2).

Reviewer #2:

*1) The paper has a very cogent Introduction pointing out the challenge of reading out static decision variables from dynamic processes. But some of the rhetoric around this topic seems to go beyond what the data support. The authors express skepticism that neural activity represents economic variables (the word "represent" frequently appears in scare-quotes). But the idea that neural signals evolve dynamically isn't incompatible with the idea that they represent economic variables. This paper's conclusions really seem to pertain to how, rather than whether, economic variables are neurally represented. The results suggest that chosen value, for example, is encoded in the LFP amplitude in a specific time window (e.g. around 400 ms where PC2 peaks, indicative of the latency of the underlying waveform), rather than, say, the overall peak of the evoked LFP.*

Both reviewers felt that we had overstated the claim that could be made with our results, and we agree with Reviewer #2 that some of our conclusions were overly rhetorical and went beyond what was supported by the data. We have sought to address both Reviewer #1 and Reviewer #2’s concerns by changing the tone and conclusions of the paper throughout.

Please see our response i)-v) above.

*2) Trial-by-trial LFP-derived indices are shown to be related to spiking activity, but it's not entirely clear how to relate this finding to what is already known about spike/LFP coupling. Could this effect be accounted for in terms of straightforward first-order correlations between the spiking and LFP time courses? What makes this seem plausible is that the time course of PC2 weights (Figure 2) looks similar to the time course of PC2 effects on spiking (Figure 3). Spikes and LFPs were recorded from different electrodes to avoid contamination, but it's still possible that a genuine relationship holds between the two (to mention just one previous example, a general negative correlation between multi-unit activity and LFP is shown in Whittingstall & Logothetis, 2009, Figure 1).*

This is an important point. A first concern is that the effects may be driven be underlying correlations between spiking and LFP, similar to the Whittingstall and Logothetis paper. We addressed this by first regressing the LFP amplitude out of the (temporally smoothed) firing rate time-course on each trial. Then we asked whether the effects shown in Figure 3 hold on the *residual* firing rate, after LFP amplitude had been regressed out.

As can be seen in Figure 3, the effect does not appear to be removed by accounting for the first-order relationship between spiking and LFP. We have added the following comment to the manuscript:

“Additional analyses confirmed that this relationship could not be explained as a simple consequence of first-order correlations between LFP amplitude and firing rate.”

However, we also suspect that the reviewer was driving at a deeper point, which is that many researchers approach spike/LFP relationships in different ways to those addressed by our paper, and it is important to draw links between this work and our own. For instance, spike-LFP coupling is often considered in terms of the relationship between spike timing relative to the phase of an oscillation. It might therefore be helpful to address how trial-wise variation in PC2 weights correlated with trial-wise variation in the phase and amplitude of underlying oscillations.

To address this, we computed a time-frequency decomposition of each trial’s LFP response. We then correlated the power and phase of these spectrograms with the underlying PC2 weights. In the case of phase, we computed the circular-linear correlation coefficient that tests the relationship between a linear variable (PC2 weights) and a circular one (phase of oscillation) (Berens P, J Stat Soft, 2009). The results of this are shown below. They indicate that, as might be expected from a shift in the latency of an evoked response, PC2 correlates with the phase of low-frequency components of the time-frequency decomposition. There was also an unpredicted relationship between PC2 weights and high beta/low gamma power relatively late into the trial. We have added Figure 2—figure supplement 3 as an additional figure supplement.

We have also added the following to the Results:

“Knowledge about PC2 weights therefore provides a parsimonious description of single-trial ERP latencies, a key feature of the dynamical ERP response. Consistent with this idea, PC2 weights were also found to correlate with the phase of theta frequency (4-–8 Hz) oscillations during the decision period (Figure 2—figure supplement 3).”

To the Discussion, we have added:

“Variation in ERP latency corresponds to a shift in the phase of an oscillation at low frequencies that make up the evoked response.[…]This suggests a way in which future studies may link our measure of LFP latency to spike-LFP coupling (Hunt, Dolan and Behrens, 2014) or inter-regional LFP coherence (Tauste Campo et al., 2015) during cognitive tasks.”

*3) I don't follow the logic of the analysis in the subsection “Chosen value correlates as an epiphenomenon of varying decision dynamics”. In a regression model of spike rates, predictors for LFP-derived PC1 and PC2 steal variance away from a "chosen value" predictor but not from other task-related predictors. This is taken to imply that "chosen value" effects are mainly picking up on the latency of the neural response. But this result actually seems consistent with these effects being related to* either *amplitude (PC1) or latency (PC2). The latency interpretation would be on stronger footing if it could be shown that chosen value effects varied with the addition/removal of PC2 only.*

We included both PC1 and PC2 as we found that they both independently could quench some of the variance in the chosen value signal (and combined together explained more). However, we can see that it would be useful to show the effects of PC2 alone. We have now added this as the reviewer suggested as Figure 4—figure supplement 2.

*4) I'm concerned about whether it's appropriate to treat LFPs from simultaneously recorded electrodes as independent observations. Panel A of Figure 4—figure supplement 2 shows that LFP features are highly correlated across electrodes, even in different brain regions. This probably isn't a problem for the PCA decomposition, but it does seem like an issue for inferential tests of associations between LFP and task variables. For example, Figure 1 plot "mean /- s.e. across electrodes"-but since multiple electrodes were recorded per trial, the nominal number of electrodes may be larger than the true number of independent observations. The same concern applies to the results in Figure 2 and related figures. It would be helpful for the paper to clarify how interdependence among electrodes was dealt with.*

This is a well thought-out concern, and one that we had not considered in our original statistical inference across recording electrodes. In practice, however, we do not believe it to be of any real practical concern, simply due to the robustness and reproducibility of the results across recording sessions (which are, of course, independent from one another). To give an example, here are the *individual electrode* Z-statistic time-series for chosen value, which comprise the findings for the DLPFC, OFC and ACC electrodes in Figure 1 and Figure 1—figure supplement 3. Each of the 40 dashed lines represents the start of a different recording session. Hence electrodes *between* each pair of dashed lines were those recorded simultaneously (Figure 9).

Author response image 2.Chosen value Z-statistics for individual electrodes across the 40 recording sessions used to create Figure 1 and Figure 1—figure supplement 3.Each dashed line reflects the beginning of a new recording session.**DOI:**
http://dx.doi.org/10.7554/eLife.11945.025

As can be seen, the effects are very reproducible across different recording sessions. But perhaps more importantly, they are not noticeably more similar/consistent *within* recording sessions than *across* recording sessions.

There are, of course, more sophisticated statistical ways of accounting for interdependence between simultaneously recorded electrodes. But such methods are not particularly widely applied in electrophysiology at present (e.g. when collapsing across simultaneously recorded neurons), and a thorough treatment of this topic might warrant a separate paper by itself. For the purposes of the present study, it wouldn’t really impact upon the conclusions we are trying to draw.

[Editors' note: the author responses to the re-review follow.]

1) The authors claim that the PC2 from LFP provides a measure of local dynamics that is more specific (and relevant to unit activity) than larger scale global dynamics. The analysis provided by the authors (effect of local LFP after orthogonalizing on LFP from another region) is encouraging in this regard, however supporting the claim would also require showing that the opposite is not true (i.e. that running the analysis in reverse does not yield the same result). The correlation coefficients in the histograms of Figure 4—figure supplement 3 suggest that this may be the case, although there does not seem to be much of a within region advantage for DLPFC in particular.

The reviewer is correct in this regard, and this point was of real concern for us in designing this local vs. global analysis. Specifically, the principal component estimates will contain observation noise on both local and distal electrodes. Because of this, running an analysis of the effect of local LFP after orthogonalising on LFP from another region may indeed be insufficient to make the claim that local activity is more specific than global dynamics.

However, due to these concerns, the analysis we originally performed in fact includes the opposite analysis, as requested by the reviewer, within it. To generate Figure 4—figure supplement 3, we first test the effect of local LFP after orthogonalising on another region; *then we run the opposite analysis; then we subtract the latter from the former*. Testing for an effect greater than zero therefore allows us to address whether local dynamics influence neuronal firing more than global firing, and more so than the opposite direction. Unless we’re misunderstanding what the reviewer is asking, we believe that this deals with the concern raised (to run the opposite analysis to the one that we’ve performed would actually just flip the y-axis of Figure 4—figure supplement 3).

We understand why the reviewer may have missed this detail of our analysis: it was buried in the figure legend of Figure 4—figure supplement 3, and not explicitly referred to in the main text. We apologise for this, and have amended the figure legend, main text and Methods as follows to make this more explicit.

Main text:

“We found a smaller but significant reduction could also still be found even if these ‘local’ principal components (i.e. from the same brain region) were orthogonalised with respect to those of another, simultaneously recorded brain region, when compared to performing the same analysis in reverse (Figure 4—figure supplement 3).”

Methods:

“For Figure 4—figure supplement 3, we sought to explore whether local LFP principal components provide greater contributions to the reduction in chosen value variance than those recorded from other areas. […]To provide a fairer comparison, we therefore performed the same analysis in reverse: examining the effect of the distal OFC/ACC PC1/2 weights, orthogonalised with respect to the local DLPFC PC1/2 weights. Figure 4—figure supplement 3 shows the latter analysis (distal orthogonalised with respect to local) subtracted from the former (local orthogonalised with respect to distal).”

Legend to Figure 4—figure supplement 3:

“Local PC1/2 (controlling for larger-scale global influences) reduces chosen value CPD more than global PC1/2 (controlling for local influences).”

*Also, as far as I can tell, the authors did not incorporate reaction time, which is typically used as a trial-by-trial indicator of decision dynamics, into these analyses. I suspect that this is because the required delay is sufficiently long to create a floor effect on RTs, however if this is the case it deserves mention in the text. Otherwise the authors should test whether LFP measures relate to the "chosen value" aspects of neural activity after accounting for RT.*

As the reviewer suspected, the required delay was indeed sufficiently long to create a floor effect on RTs. We have added a comment to the main text to acknowledge this point:

“Note that reaction time is not included as an additional measure of trial-by-trial decision dynamics in our task, as the imposed 1000 ms choice delay prior to response led to a floor effect in subjects’ response times.”

*2) The authors show that PC2 captures variance in neural firing that is typically ascribed to chosen value. I agree with the authors that this is an important point. Yet I found the interpretation of this point to be somewhat overly specific: "We therefore questioned whether chosen value coding might simply be a consequence of the same neural dynamics occurring at varying latencies across trials." This is certainly one interpretation of the authors' results, but not the only one. Another (more standard) interpretation of this finding is that the LFP components mediate the chosen value representations at the level of single units. Given that the LFPs are often interpreted as measurements of local synaptic input, I find this interpretation to be fairly reasonable and consistent with the data provided. The finding that PC1 (which does not contain timing information) also steals variance from chosen value may even provide some specific evidence for this interpretation. As I understand it, the main reason that the authors' interpret the data in terms of dynamics is provided in the final analysis showing that in an attractor network of decision-making the same sorts of signals emerge as a byproduct of competition. I think in order to improve the clarity it would be useful to avoid specific interpretations of causality until the Discussion section where both possibilities should be discussed.*

This is a balanced and fair appraisal of our findings. In response to this, and to point 3 below, we have rewritten the Results section on chosen value correlates and their relationship to PC1/2 in a more neutral tone. We agree that it makes sense to move the dynamical interpretation out of the results section as suggested. We have mentioned in the Discussion that there is more than one plausible interpretation of these findings.

In the Results section:

Previous title: “Chosen value correlates as a consequence of varying decision dynamics” has become: “Influence of single-trial LFP components on single unit chosen value correlates”.

Previous text: “We therefore questioned whether chosen value coding might simply be a consequence of the same neural dynamics occurring at varying latencies across trials” has been replaced by: “We sought to explore the relationship between chosen value correlates identified in single unit firing and our single-trial indices of ERP amplitude (PC1) and latency (PC2).”

Previous text: “This implies that chosen value coding is, at least in part, a consequence of the same choice dynamics occurring at different rates on different trials” has been removed.

In response to point 3 below, the joint importance of speed and amplitude has been emphasised: “Similar results could also be obtained […] by examining the contribution of PC1 or PC2 alone”; and “the neuronal variance attributed to chosen value correlates originates as a consequence of the speed and amplitude at which dynamics unfold.”

In the Discussion section:

Previous text: “Crucially, however, a portion of the variance captured by chosen value was then explained away by including the dynamics occurred on that trial as a coregressor, estimated via PCA decomposition of the LFP (Figure 4)” has become: “Moreover, a portion of the variance captured by chosen value was then explained away by including the speed and amplitude of the local LFP response on that trial as a coregressor, estimated via PCA decomposition of the LFP (Figure 4).”

Previous text: “As such, it may be the case that there is coding of chosen value during the decision that is not explained by the dynamics of decision formation” has become: “As such, it may be the case that there is coding of chosen value during the decision that is not explained by the amplitude and speed of the evoked response during decision formation.”

The following text has been added: “A further important caveat is that chosen value correlates may be explained by different mechanisms at different points in the trial. […] Future work may investigate the origin and functional significance of this persistent chosen value coding, perhaps for use at later task stages.”

*3) Relatedly, another reviewer noted that the most direct evidence for the notion that chosen-value effects emerge from "the same choice dynamics occurring at different rates on different trials" is derived from the analysis on p. 10, but I'm afraid I'm still not onboard with the logic of this. The authors foreground an analysis showing that the chosen-value effect in spike rates is reduced by jointly including PC1 (indexing LFP amplitude) and PC2 (indexing LFP latency) in the regression model (Figure 4, Figure 4—figure supplement 1, and Figure 4—figure supplement 3). I still don't see how this is germane to their conclusion about latency. The specific effect of the latency-related PC2 is now shown as a supplement, and is considerably weaker albeit nonzero. The original chosen-value effect, which peaked at a% CPD of around 0.9 (Figure 1), is reduced by up to about 0.1 by both PCs together (Figure 4), or 0.03 by PC2 alone (Figure 4—figure supplement 2). So wouldn't it be at least as accurate to conclude that chosen-value effects emerge from the same neural dynamics occurring at different amplitudes across trials? (The predictions of the attractor network are also unclear in this regard, i.e. in the predictions for combination of PCs, and so it seems important that the authors clarify the predictions of the attractor model.) The same concern applies to the local-versus-distal result in Figure 4—figure supplement 3, which is interpreted solely in terms of neural latency but results for PC2 alone are not shown. It seems to me an easier conclusion to square with the data would be that chosen-value effects originate from a combination of the amplitude and speed of the dynamically unfolding neural response. This may not be so starkly different from existing perspectives, although it's of course still valuable to see the details of how it plays out. The authors could also easily reframe the result that LFP measures partially mediate the chosen value effect and tone down the oppositional framing of decision dynamics versus static chosen value representation. They could use the network model results as a podium to say that chosen value representations need not emerge for the purpose of encoding chosen value, which would be entirely accurate from my standpoint.*

Thanks for these comments. In response to them (and also point 2 above), we have sought to make our results section on chosen value correlates and their relationship with PC1/2 much more balanced. We now save the interpretation for the Discussion section (i.e. once the modeling results have been introduced also), and have sought to clarify the contributions of both LFP amplitude (PC1) and latency (PC2) to the results in this section. We have made several textual edits to the Results and Discussion section, described above in response to point 2.

In addition (and in response to point 6 below), we have amended Figure 4—figure supplement 2 to now show the contribution of PC1 as well as PC2 alone, separately across all three regions.

*4) A similar issue comes up in the section on neural network modeling, which shows that chosen-value effects in the simulation can be explained away by a principal component capturing variability in neural dynamics, but doesn't state clearly enough that "dynamics" here refers, not just to latency, but to a mixture of amplitude and latency (Figure 7—figure supplement 1, panel C). (Related to this, a sentence the sentence “we obtained principal components that controlled waveform amplitude and latency, as in the ERPs" seems inaccurate, since what was actually extracted was a single component that mixed amplitude and latency, unlike the earlier analysis.)*

In the model, there is greater covariation between the amplitude and the latency of the evoked response than in the data. This is likely to be because the inputs to the network are sustained even after an attractor state has been reached. We have amended this section of the Results to reflect these details more accurately, as the reviewer suggests:

Previous text: “We then ran a similar dimensionality reduction on summed network activity, a proxy for the model’s LFP predictions, as we had done on both macaque LFP and human MEG data. […] We then regressed these principal components back onto single unit firing rates, as in Figure 3. We found a similar time-course of influence of the model’s principal components on single unit firing to that found in the data (compare Figure 3 with Figure 7, cyan).”

Now reads: “We then ran a similar dimensionality reduction on summed network activity, a proxy for the model’s LFP predictions, as we had done on both macaque LFP and human MEG data. […] We then regressed the principal component back onto single unit firing rates, as in Figure 3. We found a similar time-course of influence of the model’s principal component on single unit firing to that found in the data (compare Figure 3 with Figure 7, cyan).”

*5) The authors use the first principal component of the attractor network model decomposition rather than PC2. While I agree with the authors that this component seems to contain some temporal information, it does seem to mainly capture amplitude. Given that the authors interpret PC2 as if it were the derivative of the ERP, it seems that it would be a bit easier to make the connection between computational model and biology if model signals were decomposed using separate components to capture overall amplitude and derivative. This could be done simply by creating weight vectors based on the average model response (ERP) and the derivative of that signal.*

Thanks for this suggestion. We prefer to keep the PCA rather than introduce a different way to perform the analysis. Despite the differences in the components obtained (now discussed in response to point 4), we think it aids clarity to retain a direct relationship between our approaches to analyzing both data and the model. In the model, there is greater covariation between the amplitude and the latency of the evoked response than in the data. This is likely to be because the inputs to the network are sustained even after an attractor state has been reached.

However, the reviewer’s suggestion of using the shape of the waveform and its derivative as basis functions is potentially a helpful one for future studies. We’ve added a comment to the Discussion to highlight this idea.

In the Discussion, we now state:

“PCA is one of many possible approaches to obtaining a useful set of temporal basis functions to describe variation in ERP waveforms, and it may be improved upon by future investigations. […] This feature of the data is identified automatically using PCA.”

*6) The authors note that PC1 relates primarily to value sum, however this does not seem to be the case in ACC, where PC1 takes positive coefficients for chosen value and negative coefficients for unchosen value. This makes me wonder whether the added contributions of PC1 in explaining single unit chosen value effects relate primarily to the inclusion of PC1 from ACC electrodes. In my opinion, such a finding would not necessarily detract from the dynamics idea, as overall ACC signal amplitude may play a role in adjusting decision threshold, which directly affects dynamics. Either way, the authors should make the heterogeneity across recording locations clear in this regard.*

We have added a comment to the Results section to make clear that in ACC, PC1 has an opposite sign for chosen and unchosen value:

“PC1 weights, capturing response amplitude rather than latency, were primarily influenced by value sum (same sign for both chosen and unchosen value, with the exception of ACC)[…].”

To make clearer how the contributions PC1 and PC2 in explaining chosen value effects vary across regions, we have now extended Figure 4—figure supplement 2 to include all three regions, subdivided into PC1 and PC2 reduction in explained variance. (As noted above in response to points 2 and 3, we have also modified the main text in this section to more faithfully reflect the joint contributions of PC1 and PC2 to reducing chosen value CPD).

In ACC, the contribution of PC1 appears particularly strong, a point that we now note in the figure legend. We therefore also repeated the inter-regional analysis (Figure 5) using PC1 instead of PC2. We found that PC1 did not have any differential influence on effort vs. delay trials.

*7) The section on inter-region correlations (“OFC and ACC dynamics have distinct influences on delay- and effort-based decisions respectively”) is quite specific about causal directionality, concluding that DLPFC activity is influenced by OFC and ACC. It isn't apparent to me that there's any support for this. (A third-variable explanation would be a plausible alternative.)*

This section of the manuscript was particularly hypothesis-driven, given our knowledge about the dissociable effects of lesions to ACC and OFC (e.g. Rudebeck et al., 2006) and imaging activations (e.g. Prevost et al., 2010) in delay- and effort-based decisions. It is for this reason that we were adopted a position about the direction in which we believed that causality might arise. However, to test this more explicitly, we also performed the same analyses in reverse (using DLPFC/OFC to explain ACC firing, and DLPFC/ACC to explain OFC firing). As shown below (Figure 10), neither of these analyses produced significant effects.

Author response image 3.The same analysis as in main Figure 5, repeated with ACC firing being the dependent variable (left) and OFC firing being the dependent variable (right).**DOI:**
http://dx.doi.org/10.7554/eLife.11945.026

We now note these two null results in the main text:

“Both areas were found to influence DLPFC activity (Figure 5—figure supplement 1), but strikingly, ACC explained more variance in DLPFC firing on effort trials than delay trials (Figure 5, magenta), whereas the converse was true for OFC (Figure 5, black). This was not found to be true for analyses performed in the reverse direction (using DLPFC/ACC PC2 weights to explain OFC firing, or DLPFC/OFC PC2 weights to explain ACC firing).”

However, we agree with the reviewer that there is always the possibility of a third-variable explanation whenever considering functional connectivity analyses between two regions. We have amended the Discussion to acknowledge this point:

“We note, however, that our analysis does not test the possibility is that a third (unobserved) variable could jointly affect both regions, doing so differentially on effort vs. delay trials.”